# Secretion of protein disulphide isomerase AGR2 confers tumorigenic properties

Delphine Fessart[1,2,3,4*], Charlotte Domblides[5], Tony Avril[1,2], Leif A Eriksson[6], Hugues Begueret[7], Raphael Pineau[8], Camille Malrieux[3,4,5], Nathalie Dugot-Senant[9], Carlo Lucchesi[4,10], Eric Chevet[1,2†], Frederic Delom[3,4,5†]

[1]Oncogenesis, Stress and Signaling Laboratory, ERL440 Inserm, Université de Rennes 1, Rennes, France; [2]Centre de Lutte Contre le Cancer Eugène Marquis, Rennes, France; [3]INSERM U1218, Actions for onCogenesis understanding and Target Identification in ONcology (ACTION), Bordeaux, France; [4]Bergonié Cancer Institute, Bordeaux, France; [5]Université de Bordeaux, Bordeaux, France; [6]Department of Chemistry and Molecular Biology, University of Gothenburg, Göteborg, Sweden; [7]Hôpital Haut-Lévêque, Bordeaux, France; [8]Animalerie mutualisée, Université de Bordeaux, Bordeaux, France; [9]Université de Bordeaux, Bordeaux, France; [10]Site de Recherche Intégrée sur le Cancer, Bordeaux Recherche Intégrée en Oncologie, Bordeaux, France

*For correspondence: delphine. fessart@yahoo.fr

†These authors contributed equally to this work

Competing interests: The authors declare that no competing interests exist.

**Abstract** The extracellular matrix (ECM) plays an instrumental role in determining the spatial orientation of epithelial polarity and the formation of lumens in glandular tissues during morphogenesis. Here, we show that the Endoplasmic Reticulum (ER)-resident protein anterior gradient-2 (AGR2), a soluble protein-disulfide isomerase involved in ER protein folding and quality control, is secreted and interacts with the ECM. Extracellular AGR2 (eAGR2) is a microenvironmental regulator of epithelial tissue architecture, which plays a role in the preneoplastic phenotype and contributes to epithelial tumorigenicity. Indeed, eAGR2, is secreted as a functionally active protein independently of its thioredoxin-like domain (CXXS) and of its ER-retention domain (KTEL), and is sufficient, by itself, to promote the acquisition of invasive and metastatic features. Therefore, we conclude that eAGR2 plays an extracellular role independent of its ER function and we elucidate this gain-of-function as a novel and unexpected critical ECM microenvironmental pro-oncogenic regulator of epithelial morphogenesis and tumorigenesis.

## Introduction

It is well established that proper organization, maintenance, and function of the epithelial organs are largely ascribed to cell-cell adhesion and polarization of epithelium and its interaction with extracellular matrix (ECM). Apical-basal polarity is a fundamental feature of epithelial cells required for cell division, differentiation and morphogenesis (*Arumugam et al., 2008*; *Nelson, 2009*). Polarization is initiated by signaling from cell-to-cell and cell-to-ECM contacts (*Nelson and Bissell, 2006*). Polarization of epithelial cells is implicit for the development of lumens, which are essential for glandular tissues to carry out their normal functions. Conversely, loss of tissue organization is among the earliest hallmarks of epithelial cancer. Therefore, understanding mechanisms of lumen and apical surface formation will improve future prospects for therapy.

*Anterior gradient 2 (AGR2)* encodes an endoplasmic reticulum (ER)-resident protein mainly expressed in epithelial cells in human. Enhanced intracellular AGR2 (iAGR2) expression is observed in many cancers (reviewed in Ref [*Chevet et al., 2013*]). Previously, we have demonstrated that

**eLife digest** Cancer cells multiply abnormally fast and therefore produce protein molecules faster than normal cells. To avoid becoming stressed by this overproduction, cancer cells make use of proteins that fold the new proteins inside the cell. One of these protein folders is called anterior gradient-2 (or AGR2 for short) and is produced at high levels in so-called epithelial cancers, such as breast and lung cancer.

Previous research has shown that AGR2 inside cancer cells can help them grow and survive and AGR2 can also be found outside cells, such as in the blood or the urine of cancer patients. Therefore some researchers have suggested that measuring the levels of AGR2 in bodily fluids may be a useful marker for detecting cancers. Fessart et al. hypothesized that – apart from becoming a promising diagnostic tool – the AGR2 protein itself, specifically when found outside cells, might make cancer cells more aggressive.

Fessart et al. used a range of techniques to test this hypothesis. For example, healthy lung cells and lung cancer cells were grown into miniature replicas of lung organs in the laboratory, and in a key experiment, AGR2 was added to the lung organoids grown from the healthy cells. The addition of AGR2 protein was enough to change the non-tumor organoids into tumor organoids and boosted their growth about ten-fold. Further experiments then revealed that AGR2 also makes cells more invasive and capable of moving, both important features of aggressive cancer cells.

Overall, Fessart et al. have proven that AGR2 is a signalling molecule found outside cancer cells that makes them more aggressive. In future, more research addressing how AGR2 achieves this may lead to new therapeutic strategies against some forms of cancer.

iAGR2 overexpression could represent a mechanistic intermediate between endoplasmic reticulum quality control (ERQC) and tumor development (*Higa et al., 2011*; *Chevet et al., 2013*). In such model, increased iAGR2 expression could enhance ER protein homeostasis/proteostasis thereby allowing tumor cells to cope with abnormal protein production and secretion and contributing to the aggressiveness of cancer (*Higa et al., 2011*). The latter was demonstrated using both in vitro and in vivo approaches (*Chevet et al., 2013*). Although the iAGR2-mediated ER proteostasis control model is appealing, it was also observed that in cancer, AGR2 was present in the extracellular space, serum, and urine (*Shi et al., 2014*; *Park et al., 2011*), thereby opening other avenues for its role on tumor microenvironment. Despite the detailed characterization of its intracellular function, the physiological role of extracellular AGR2 (eAGR2) remains unknown. AGR2 is a Protein-Disulfide Isomerase (PDI), PDIA17 (*Persson et al., 2005*), and although the intracellular roles of PDIs have been well documented, some of these proteins were also found in the extracellular milieu, with unclear functions. For instance, we have previously shown that PDIA2 is secreted into the lumen of the thyroid follicles by thyrocytes to control extracellular thyroglobulin folding and multimerisation (*Delom et al., 1999*; *Delom et al., 2001*). Further, PDIA3 was found to be secreted and to interact with ECM proteins (*Dihazi et al., 2013*) and QSOX1 was reported to participate in laminin assembly thereby controlling ECM functionality (*Ilani et al., 2013*).

We and others, have recently demonstrated that epithelial organization and many physiological cell-cell and cell-ECM contacts, cellular polarity, and secretory functions are preserved in epithelial organoids (*Fessart et al., 2013*; *Kimlin et al., 2013*). Therefore, to address whether eAGR2 could act as a pro-oncogenic molecule in the ECM, we have used our human epithelial organoid model (*Fessart et al., 2013*). We demonstrate, for the first time, that eAGR2 plays an extracellular role independent of its ER function and we elucidate this gain-of-function as a novel and unexpected critical ECM microenvironmental pro-oncogenic regulator of epithelial morphogenesis and tumorigenesis.

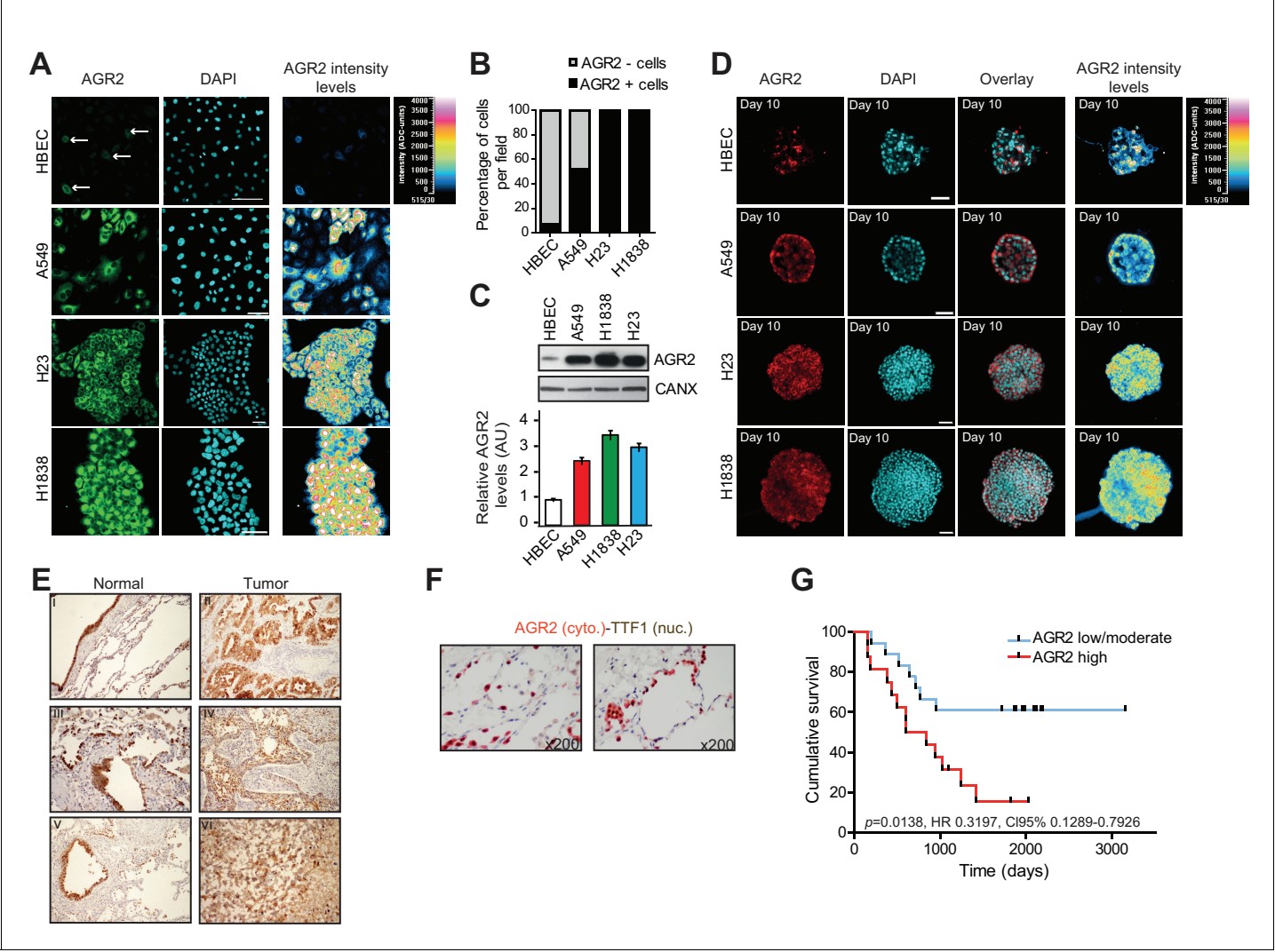

**Figure 1.** AGR2 is overexpressed in lung cancer cell lines and tumor tissues. (**A**) Analysis by immunofluorescence of AGR2 expression in normal human bronchial epithelial cells (HBEC) and three lung cancer cell lines (A549, H23 and H1838) grown in 2D culture. Scale bars, 50 μm. (**B**) Quantification of AGR2 protein expression in cell lines according to immunofluorescence. The stacked bars show the percent contribution of high and low AGR2-positive cells relative to the total number of cells per field. (**C**) Expression of AGR2 protein detected by Western blot in a panel of human lung epithelial cell lines. Values correspond to three independent experiments. Data are mean ± SEM. (**D**) Confocal cross-sections of organoids stained with AGR2 antibody (red) and DAPI (blue) for nucleus, in normal HBECs and three lung cancer cell lines (A549, H23 and H1838). Scale bars, 50 μm. (**E**) AGR2 expression determined by immunohistochemistry in sections of formalin-fixed paraffin-embedded normal human lung samples and in the different lung adenocarcinoma subtypes. II = squamous cell carcinoma, IV = adenocarcinoma, VI = large cell carcinoma as compared to normal tissues (I,III,V) (x200). (**F**) Pulmonary lung carcinoma showing brown, nuclear immunostaining for TTF-1 expression and cytoplasmic immunostaining for AGR2 expression (dual color Multiplex TTF-1 + AGR2 immunostain; x200). (**G**) Kaplan-Meier survival curves of lung cancer patients. The cumulative survival was related to different levels of AGR2 expression: Group 1, low to moderately positive stains (n=17); and Group 2, strongly positive AGR2 stains (n=16), as defined in Materials and methods and *Supplementary file 1A*.

## Results

### AGR2 overexpression in human lung adenocarcinoma correlates with poor clinical outcome

To evaluate the correlation between AGR2 expression levels and lung cancer, we monitored AGR2 endogenous expression in a panel of human lung bronchial epithelial cell lines. High AGR2 expression was only observed in lung tumor cell lines (A549, H23, H1838) compared to a non-tumorigenic human bronchial epithelial cell (HBEC) (*Figure 1A–C*). Moreover, the expression pattern of AGR2 in

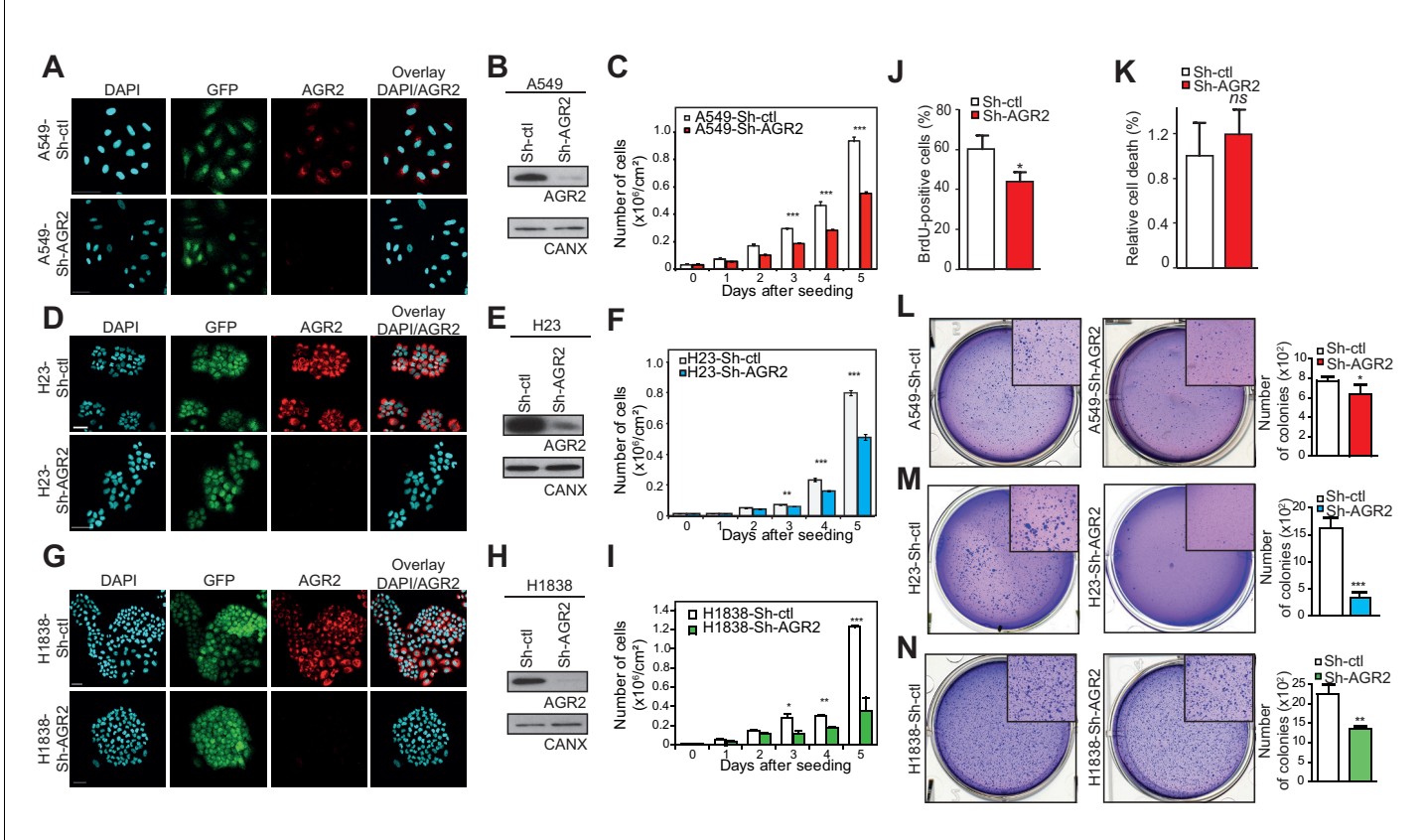

**Figure 2.** AGR2 knock down in lung cancer cells decreases cell proliferation and malignant transformation. (A, D, G) Analysis by immunofluorescence of AGR2 depletion in our three lung cancer cell lines (A549, H23, and H1838). Scale bars, 50 μm. (B, E, H) Western blot analysis showing the down-regulation of AGR2 protein levels in control and Sh-AGR2 transfected cells. Calnexin (CANX) concentrations are shown as the loading control. One representative experiment (n = 3) is shown. (C, F, I) Growth of cells harboring Sh-ctl or Sh-AGR2 (three independent experiments). Data are mean ± SEM. (J) Quantification of the percentage of BrdU positive cells. Data are presented as mean ± SEM of at least three independent experiments. *P < 0.05. (K) Quantitation of cell death in AGR2 depleted cells (Sh-AGR2). Results are representative of three independent experiments. The cell death rate was determined by Trypan blue dye-exclusion assay. *n.s:* not significant. (L–N) Soft agar colony formation of cells expressing Sh-ctl or Sh-AGR2. The graph shows the number of colonies (mean ± SEM.) after 3 weeks of three independent experiments. The p values (determined by Student's t test) are relative to Sh-ctl cells. *p≤0.05. **p≤0.01 and ***p≤0.001. Shown at the left are representative images of the colonies formed by each cell type.

The following figure supplement is available for figure 2:

**Figure supplement 1.** Cell proliferation on A549-Sh-AGR2 depleted cells.

tumor and non-tumor bronchial organoids (*Figure 1D*) was similar to that observed in 2D culture (*Figure 1A*). Immunohistochemistry of AGR2 in a cohort of 34 non-small cell lung cancer (NSCLC) patients (*Supplementary file 1A*) revealed that AGR2 was overexpressed in tumors compared to adjacent non-tumor tissue (*Figure 1E*). Consequently, AGR2 expression was increased in NSCLC tissues (*Figure 1E*), and was essentially restricted to type II pneumocytes (*Figure 1F*). We then used a log-rank test with Kaplan–Meier estimates to analyze the cohort in order to stratify patient samples as having high, low/intermediate AGR2 expression status (*Supplementary file 1A*). High AGR2 expression correlated with low survival rate and the low/intermediate AGR2 expression with high survival rate in NSCLCs patients (*Figure 1G*). Hence NSCLC patients can be sorted into poor and good prognosis groups as a function of high or low/intermediate AGR2 expression levels, respectively. Taken together, these results demonstrate in vitro and in vivo correlations between AGR2 expression levels and lung cancer, suggesting a function for this protein in tumor development, progression and aggressiveness.

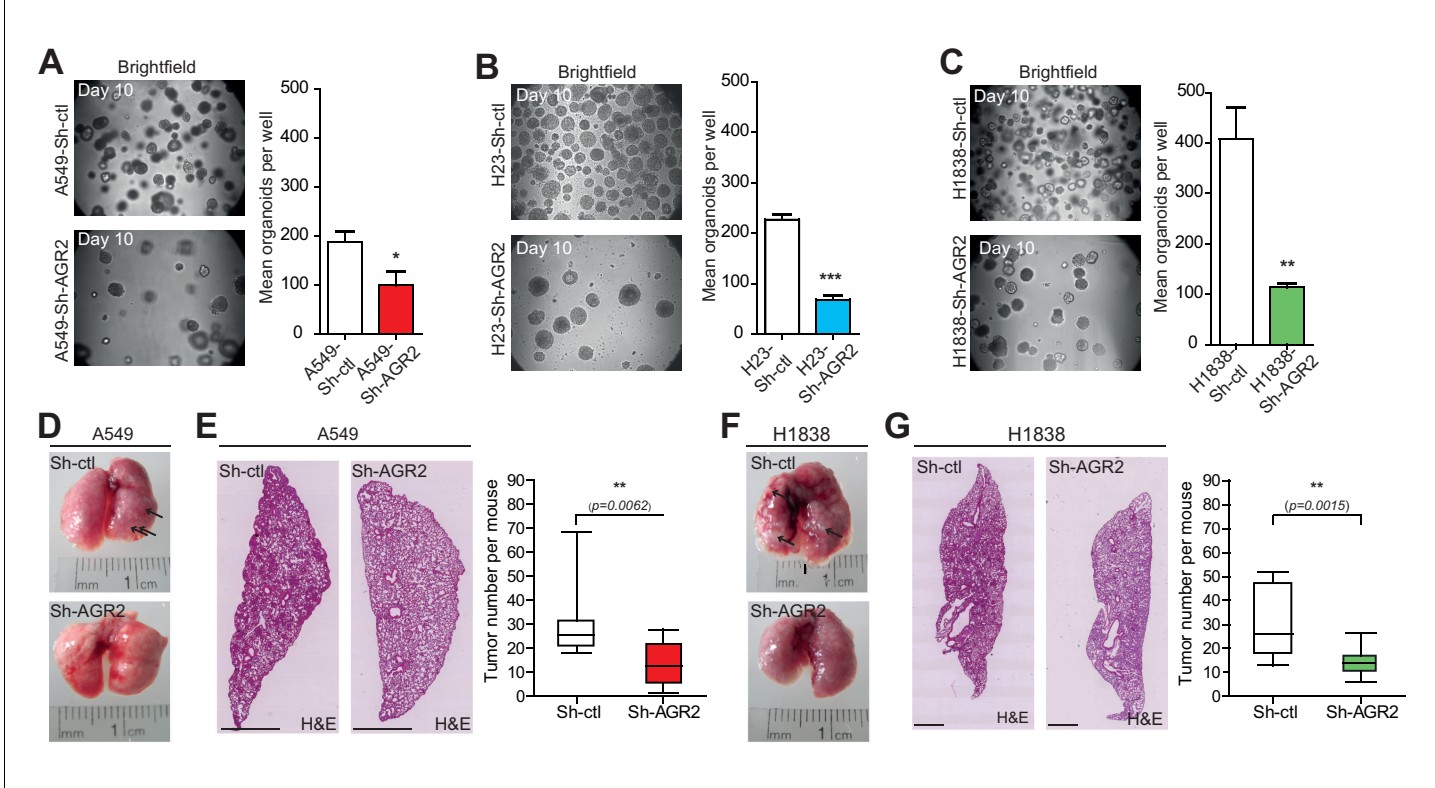

**Figure 3.** AGR2 knockdown reduces tumorigenicity and metastatic potential. (**A–C**) Representative brightfield pictures of tumor organoids grown in 3D from cells harboring Sh-ctl or Sh-AGR2 (three independent experiments). The bar graph shows the mean of organoids per well (mean ± SEM.) after 10 days of culture from three independent experiments. The p values (determined by Student's t test) are relative to Sh-ctl cells. *p≤0.05. **p≤0.01 and ***p≤0.001. (**D, F**) Representative pictures of the lungs in mice injected with Sh-ctl cells or Sh-AGR2 cells allowed to developed tumors for 2 weeks. Note the extensive lesions (white areas) on the surface of lungs from Sh-ctl mice and the relatively normal appearance of the lungs from Sh-AGR2 mice. (**E, G**) Representative H&E-stained sections for lungs injected with Sh-ctl cells or Sh-AGR2 cells (montage of x5 magnification of lung sections). Scale bars, 1 mm. Box plot showing the mean number of tumors per mouse in Sh-ctl (n=6) vs Sh-AGR2 (n=6) groups. The p values (determined by Student's t test) are relative to Sh-ctl cells. **p≤0.01.

## Loss of tumorigenicity and metastatic potential in AGR2-depleted lung adenocarcinoma cells

To test the relevance of AGR2 overexpression in tumorigenesis, we silenced its expression in lung adenocarcinoma cell lines (A549, H23 and H1838) using lentivirus-mediated infection with AGR2 shRNA (Sh-AGR2). First, we monitored the efficacy of AGR2 depletion using immunofluorescence (*Figure 2A,D,G*) and Western blotting (*Figure 2B,E,H*). Cell growth was strongly inhibited in Sh-AGR2 cells compared to control cells (Sh-ctl) (*Figure 2C,F,I–J* and *Figure 2—figure supplement 1*) with no cell death detected (*Figure 2K*). Interestingly, soft-agar colony formation (*Figure 2L–N*) (used as indicator of malignant transformation) significantly reduced upon AGR2 silencing in adeno-carcinoma cell lines, thereby indicating that AGR2 may favor tumor progression by increasing anchorage-independent growth. To further characterize the role of AGR2 in cancer, we assessed the impact of AGR2 depletion using the 3D human bronchial organoids system which we previously established (*Fessart et al., 2013*). AGR2 silencing dramatically decreased the formation of tumor organoids (*Figure 3A–C*). To determine whether AGR2 is required for tumor progression in vivo, we then evaluated the in vivo tumorigenic potential of Sh-ctl and Sh-AGR2 cells using two different ade-nocarcinoma cell lines (A549 and H1838), following tail vein injection in immunodeficient mice. Sh-ctl mice presented large metastases that were easily detected at the surface of the lungs compared to mice injected with Sh-AGR2 cells (*Figure 3D,F*). Moreover, AGR2 depletion significantly reduced total metastasis number (by ~50%) in both A549-Sh-AGR2 and H1838-Sh-AGR2 injected mice

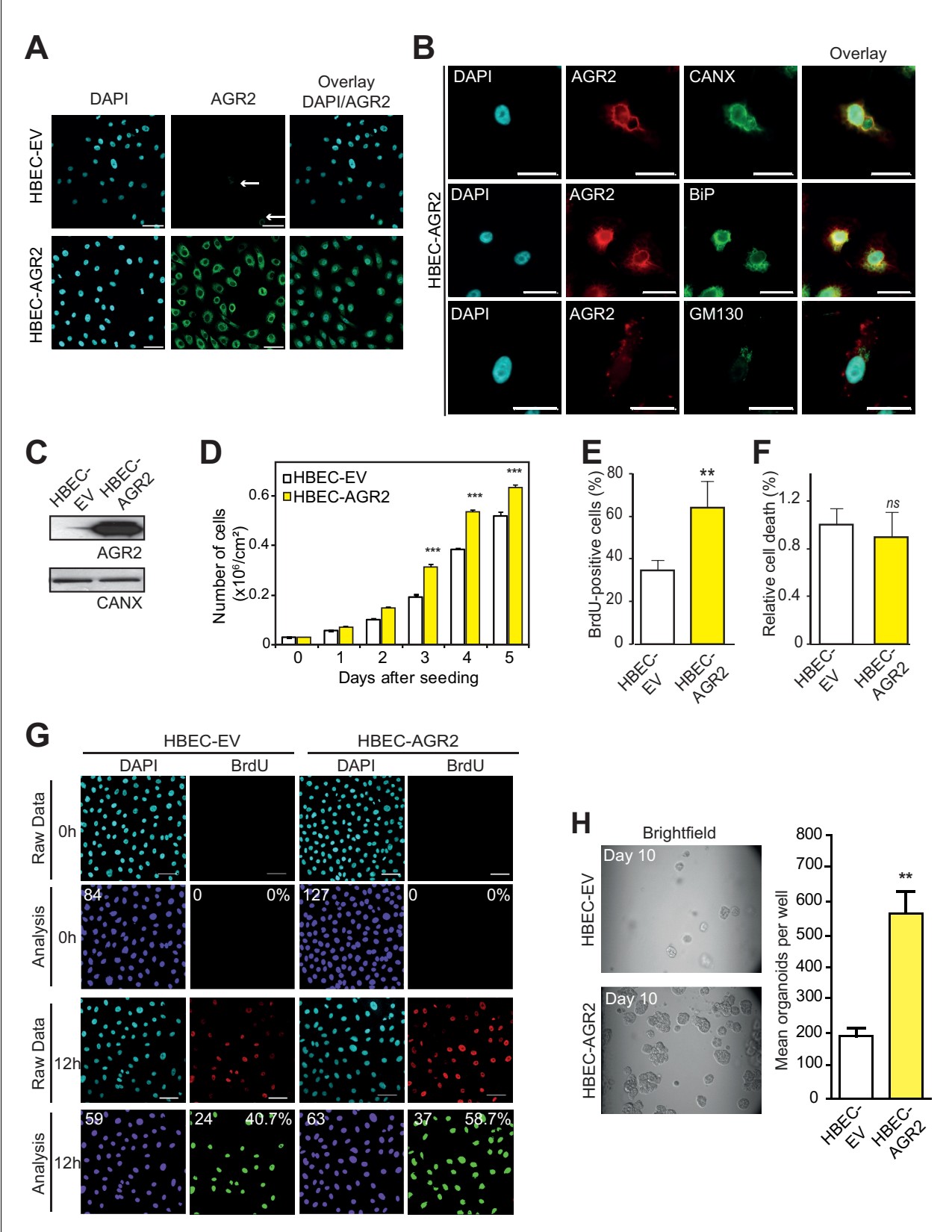

**Figure 4.** Enhanced cell proliferation and transformation after AGR2 overexpression in HBECs. (**A**) Analysis by immunofluorescence of AGR2 overexpression in HBECs. Scale bars, 50 µm. (**B**) Analysis of AGR2 sub-cellular localization in infected HEBCs by immunofluorescence. Calnexin (CANX)

*Figure 4 continued on next page*

Figure 4 continued

and Bip (GRP78) are used as an ER localization control and GM130 is used as a Golgi localization control. Scale bars, 50 µm. (C) Up-regulation of AGR2 protein concentrations in control cells (HBEC-EV) and in cells infected with AGR2 (HBEC-AGR2), as analyzed by western blot. Calnexin (CANX) concentrations are shown as the loading control. One representative experiment (n = 3) is shown. (D) Growth of cells stably expressing AGR2 or empty vector (EV) (three independent experiments). Data are mean ± SEM. (E) Quantification of the percentage of BrdU positive cells. Data are presented as mean ± SEM of at least three independent experiments. **p<0.001. (F) Quantitation of cell death in AGR2 overexpression cells. Results are representative of three independent experiments. The cell death rate was determined by Trypan blue dye-exclusion assay. n.s: not significant. (G) Cell proliferation on HBEC overexpressing AGR2 cells. Representative images of control HBECs (HBEC-EV) and HBEC-AGR2 stained for BrdU (red) administrated 12 hr before, and with DAPI (blue). Images were subjected to high throuput imaging (bottom panels, analysis data). After acquisition of the dataset, images were segmented by a watershed transformation to identify individual cells (bottom right). Scale bars: 100 µm. (H) The bar graph shows the mean of organoids per well (mean ± SEM.) after 10 days of culture from three independent experiments. Shown at the left are representative images of the organoids formed by each type of cell HBEC-AGR2 and HBEC-empty vector. The p values (determined by Student's t test) are relative to empty vector infected cells. **p≤0.01 and ***p≤0.001.

(*Figure 3E,G*). Taken together, these results show that overexpression of AGR2 in cancer cells increases their tumorigenic and metastatic potential.

## Overexpression of AGR2 increases cell proliferation

To further confirm that AGR2 is essential for cancer tumorigenesis, we next investigated the effects of overexpressing AGR2 in non-tumorigenic HBECs using lentivirus-mediated infection with either empty vector (HBEC-EV) (*Figure 4A*, top panels) or AGR2 containing vector (HBEC-AGR2) (*Figure 4A*, bottom panels). In HBEC-AGR2 cells, AGR2 co-localized with calnexin (CANX), an ER resident type I integral membrane protein (*Figure 4B*, top panels) and BiP (GRP78), a soluble HSP70 molecular chaperone located in the lumen of the ER (*Figure 4B*, middle panels). In contrast and as expected, AGR2 did not co-localize with GM130, a cis-Golgi marker (*Figure 4B*, bottom panels). These results show that AGR2 localized to the ER in HBEC-AGR2 cells (*Figure 4B*). Increased AGR2 expression in these cells (HBEC-AGR2) was also confirmed using Western blotting (*Figure 4C*). Remarkably, AGR2 overexpression enhanced cell proliferation (*Figure 4D–G*) and increased the formation of organoids (*Figure 4H*), thereby indicating that the expression of AGR2 dictates organoid-initiating frequency.

## AGR2 is secreted in the extracellular medium

Since AGR2 has been detected in the body fluids from cancer patients (*Chevet et al., 2013*) as well as in the conditioned media from pancreatic and prostate cancer cells (*Arumugam et al., 2008*; *Zhang et al., 2005*), we hypothesized that tumor organoids may also secrete AGR2 (extracellular AGR2 / eAGR2). Indeed, eAGR2 was detected in the extracellular medium of adenocarcinoma organoids (Sh-ctl), but not in adenocarcinoma organoids silenced for AGR2 (Sh-AGR2) (*Figure 5A*). To ensure that the presence of eAGR2 in the extracellular medium correlated to its overall level of expression, we also assessed the levels of eAGR2 in non-tumor organoids overexpressing AGR2 (HBEC-AGR2) (*Figure 5B*). Similarly, we found that eAGR2 was secreted in the extracellular medium from HEBC-AGR2 organoids (*Figure 5B*). We also tested the presence of two other proteins of the secretory pathway: Calnexin (CANX) and BiP/GRP78 (*Figure 5B*). None of these ER resident proteins were detected in the extracellular medium (*Figure 5B*). This demonstrates the specific presence of ER-resident AGR2 in extracellular medium from non-tumor organoids overexpressing AGR2 (HBEC-AGR2).

To further characterize the extent of AGR2 secretion, a ratiometric comparison between intracellular and extracellular AGR2 proteins was performed (*Figure 5C–D*). AGR2 protein was immunoprecipitated from both intracellular and extracellular organoid extracts and analyzed by Western blotting (*Figure 5C*). In adenocarcinoma organoids (H23), we found ~20% of eAGR2 and ~80% of iAGR2 (*Figure 5D*), while in non-tumor organoids (HBEC-EV), AGR2 protein was exclusively intracellular (*Figure 5D*). In contrast, AGR2 overexpression in non-tumor organoids (HBEC-AGR2) forced the secretion of eAGR2 (~30% in the extracellular medium) (*Figure 5D*). Next, to test if the distribution of AGR2 could modify other components of the secretory pathway, a ratiometric comparison of both BiP and CANX was performed in tumor organoids (Sh-ctl) as compared to tumor organoids silenced for AGR2 (Sh-AGR2) (*Figure 5E* and *Figure 5—figure supplement 1A*). Results showed

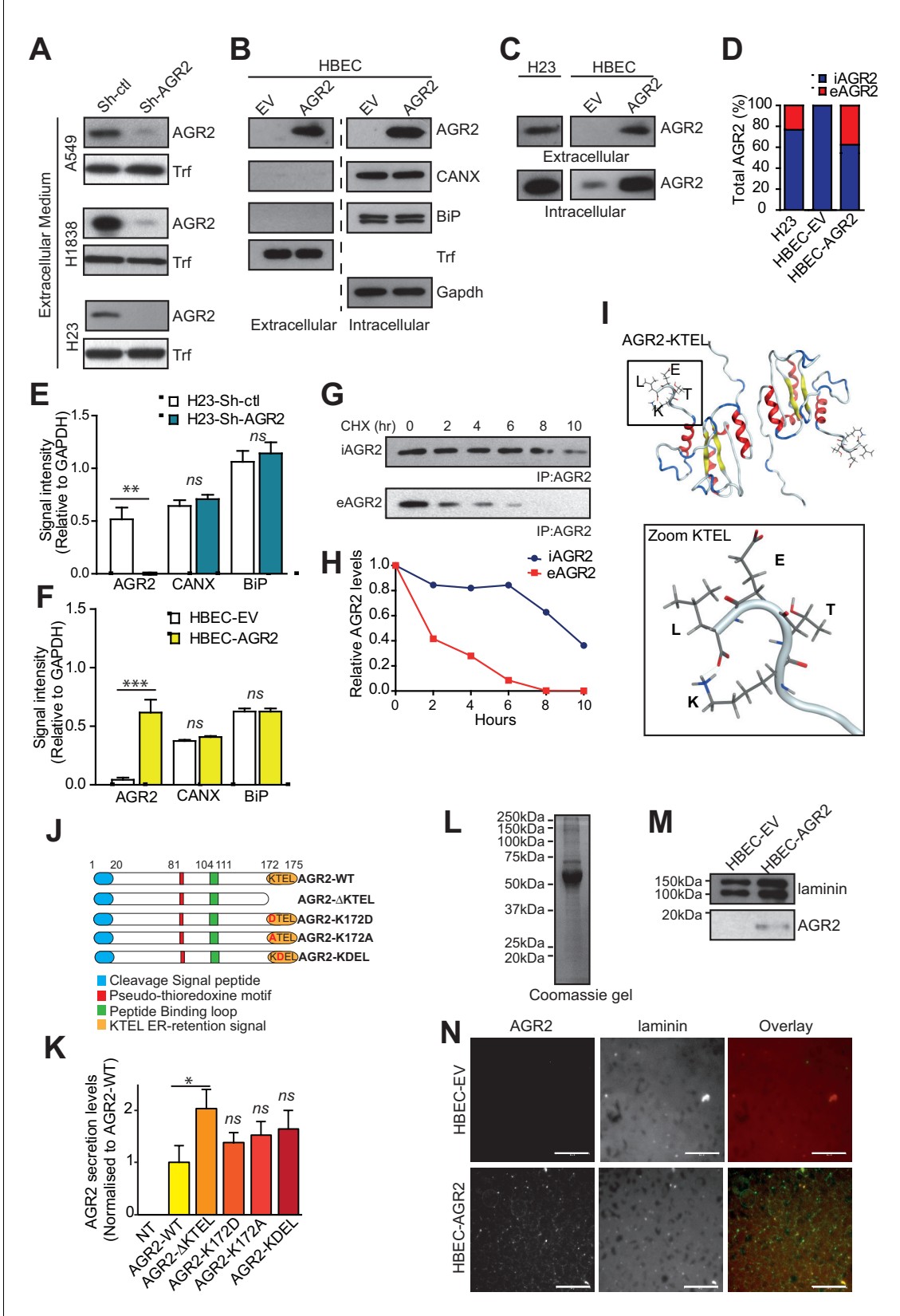

**Figure 5.** AGR2 is secreted in the extracellular medium and interacts with extracellular matrix (ECM). (**A**) Proteins in the conditioned medium were subjected to immunoblotting with anti-AGR2 antibody. Transferrin (Trf) concentrations are shown as the loading control. One representative experiment

*Figure 5 continued on next page*

Figure 5 continued

(n = 3) is shown. (**B**) Up-regulation of AGR2 protein concentrations in extracellular and intracellular lysates of HBEC infected with either empty vector (HBEC-EV) or AGR2 (HBEC-AGR2), as analyzed by western blot. Trf concentrations are shown as the loading control for extracellular lysates and GAPDH concentrations are shown as loading control for intracellular lysates. The transmembrane protein Calnexin (CANX) and the KDEL containing protein BiP are shown as control ER-resident proteins in extracellular and intracellular lysates. One representative experiment (n = 3) is shown. (**C**) Levels of extracellular and intracellular AGR2 protein detected by immunoblotting in H23 organoids as compared to the levels of extracellular and intracellular AGR2 protein in HBEC organoids expressing (HBEC-AGR2) or not (HBEC-EV) AGR2. One representative experiment (n = 3) is shown. (**D**) The stacked bars show the relative percentage of intracellular vs extracellular AGR2. (**E,F**) Quantification of the relative signal intensity of AGR2 and two other ER-resident proteins (CANX and Bip) compared to GAPDH in intracellular lysates from AGR2-depleted and control H23 organoids (**E**) or from AGR2 overexpressing and control HBEC organoids (**F**). (**G**) H23 cells were treated with CHX (cycloheximide) for the indicated periods (hours) before cell lysates and extracellular medium were prepared as described in Materials and methods. AGR2 was immunoprecipitated in intracellular and extracellular lysates with a monoclonal AGR2-specific antibody and then detected by immunoblot analysis. (**H**) The densitometric quantification of the remaining protein expressed as relative to the starting amount is shown in the diagram. (**I**) Structure model of the KTEL motif (K172-T173-E174-L175) of AGR2, PDB ID 2LNS. The four residues (KTEL) of the ER retention motif are shown in the zoom (bottom panel). (**J**) Schematic representation of the AGR2 point or deletion mutants (red: thioredoxin-like domain; orange: ER-retention motif). (**K**) Impact of the thioredoxin-like domain and of the ER-retention motif on AGR2 secretion. Quantification of the relative amount of secreted AGR2 for all the mutants. AGR2 secretion was normalized in each experiment and is compared to control non-transfected HEK-293T cells (NT). Data are mean ± SEM. *p<0.05. (**L–M**) Cell-derived matrices were generated from HBEC-EV and HBEC-AGR2 cells after 8 days of culture. Proteins were solubilized, separated by SDS-PAGE and stained with Coomassie Blue (**L**) or immunoblotted for Laminin or AGR2 (**M**). (**N**) Cell-derived matrices from HBEC-EV and HBEC-AGR2 cells were processed for immunofluorescence, stained with anti-laminin (red) and anti-AGR2 antibodies (green) and images were collected by microscopy. Scale bars: 100 μm.
The following figure supplement is available for figure 5:

**Figure supplement 1.** AGR2 expression does not alter the secretory pathway and AGR2 secreted is not O-GlcNacylated.

that of AGR2 depletion had no impact on BiP and CANX expression. Similarly, we found no difference in the expression of BiP and CANX between non-tumor organoids overexpressing AGR2 (HBEC-AGR2) or not (HBEC-EV) (*Figure 5F* and *Figure 5—figure supplement 1B*) indicating that the secretory pathway was not altered upon changes in AGR2 expression. To further correlate the amount of AGR2 produced by tumor organoids to the capacity of cancer cells to secrete eAGR2, we transiently blocked protein synthesis using a cycloheximide (CHX) pulse-chase approach and evaluated the amounts of iAGR2 and eAGR2 in tumor organoids. After 8 hr of CHX treatment, eAGR2 was practically undetectable (*Figure 5G–H*), whereas ~60% of iAGR2 was still present. These results demonstrate that CHX treatment significantly reduces AGR2 protein synthesis which impacts on AGR2 protein secretion. Indeed, extracellular AGR2 rapidly disappeared from the culture medium through yet unknown mechanisms and became undetectable within 8 hr of CHX treatment, due to the attenuated iAGR2 synthesis and the subsequent reduction of the secreted pool. Although we have not yet identified the mechanisms by which eAGR2 disappears from the medium, several hypothesis could be postulated concerning its extracellular fate. As such the secreted protein could be i) trapped in the ECM and become unavailable as a free protein in the medium; ii) aggregated and therefore remain in the unsoluble fraction before biochemical analysis; iii) degraded by extracellular proteases released by the cell or even iv) internalized by the cells and degraded through an endolysosomal pathway. Thus, the precise mechanisms by which eAGR2 disappears remain to be further investigated.

Recently, AGR2 was demonstrated to be O-glycosylated upon secretion (*Clarke et al., 2015*). Protein O-GlcNAcylation was examined in non-tumor organoids overexpressing AGR2 (HBEC-AGR2) or not (HBEC-EV) following eAGR2 immunoprecipitation (*Figure 5—figure supplement 1C*). In intracellular organoid extracts, similar O-GlcNAc levels were observed between HBEC-EV and HEBC-AGR2 organoids (*Figure 5—figure supplement 1C*) and, as expected, the presence of AGR2 was only detected in HEBC-AGR2 organoids (*Figure 5—figure supplement 1C*). In the extracellular medium from HEBC-AGR2 organoids, eAGR2 was detected but it was not O-glycosylated (*Figure 5—figure supplement 1C*).

In our experimental system, we found that in contrast to BiP, AGR2 was secreted in the extracellular medium (*Figure 5A–B*). This questioned the accessibility of the ER-retention motif present on AGR2, namely the KTEL sequence. Although AGR2 KTEL was described to bind to KDELR1 and to a lesser extent to KDELR2 and 3 (*Alanen et al., 2007*), the secretion of AGR2 might indicate a

mechanism specific to this protein. Structural analysis of AGR2 showed that the KTEL motif stemmed out of the core protein and had no interaction with the rest of the protein. The protein backbone is displayed as 'tube' model with the atoms in the KTEL sequence shown in stick format (*Figure 5I*). Residue interactions within the KTEL motif are seen more clearly: the $NH_3^+$ group of K172 forms a very clear salt bridge to the C-terminal carboxylic group of L175, which yields a stable 'hook' shape (*Figure 5I*, bottom panel). Thus, it appears that the KTEL motif really stands out of the core protein and is accessible to bind to its receptors. To explore the functional significance of the KTEL motif for the AGR2 secretion, HEK-293T cells which do not secrete AGR2, were used (*Figure 5J–K*). HEK-293T cells were transfected with different AGR2 mutant constructs (*Figure 5J*) in which the KTEL motif was mutated to either KDEL, K172D, K172A or a STOP was inserted before the KTEL (ΔKTEL) and the secretion of AGR2 protein was assessed by the presence of AGR2 protein in the extracellular medium. The eAGR2 levels in cells expressing the AGR2 mutants, in the extracellular milieu, were at least equal to that observed in cells expressing wild-type AGR2. Therefore, none of our different KTEL motif mutants displayed a specific intracellular localization and all were secreted.

Collectively, our results demonstrate that eAGR2 protein is secreted in the extracellular medium, independently of its KTEL motif and most likely as a soluble protein (eAGR2). AGR2 secretion results from high iAGR2 expression levels and is not a consequence of cell death or 'leak' of ER proteins into the extracellular medium.

## The secreted eAGR2 interacts with *extracellular matrix* (ECM)

Several secreted PDIs were shown to interact with ECM proteins (*Dihazi et al., 2013*; *Ilani et al., 2013*). Therefore, we hypothesized, in a similar manner, that eAGR2 could interact with the ECM. To investigate the interaction between ECM and eAGR2, HBEC cell-derived ECM was purified from HBEC cells overexpressing AGR2 (HBEC-AGR2) or not (HBEC-EV). Since the synthesis and organization of ECM increases with time (*Rashid et al., 2012*), ECM was isolated after 8 days of culture (*Figure 5L*). Western blotting revealed the presence of the ECM component laminin in both HBEC-EV and HBEC-AGR2 cell-derived matrices (*Figure 5M*), however AGR2 was only detected in the ECM derived from HBEC-AGR2 cell (*Figure 5M*). To further confirm that eAGR2 interacts with the ECM, immunofluorescence analysis of ECMs from HBEC-EV and HBEC-AGR2 cells was performed. ECM from both cell types stained positive for laminin (*Figure 5N*). As expected, immunofluorescence staining for AGR2 revealed the presence of eAGR2 in the ECM of HBEC-AGR2 cells (*Figure 5N*, lower panels). These results demonstrate the presence of eAGR2 in the ECM.

## Extracellular AGR2 has a physiological function in epithelial organoid formation

Given that eAGR2 is secreted by tumor organoids (*Figure 5A*) and that AGR2 has been reported in the serum of cancer patients with an average level of eAGR2 ranging from 0.5 to 20 ng/ml (*Chen et al., 2010*), we assessed whether eAGR2 might exhibit specific extracellular functions. To this end, we used different adenocarcinoma cell lines depleted for AGR2 (A549-Sh-AGR2, H23-Sh-AGR2 and H1838-Sh-AGR2). Remarkably, the addition of recombinant human AGR2 (40 ng/ml) (*Figure 6—figure supplement 1A–F*) to the medium of AGR2-depleted tumor cell lines reversed the cell growth inhibition induced by AGR2 depletion (*Figure 6A*). Moreover, the addition of eAGR2 to the ECM of AGR2-depleted organoids restored the formation of tumor organoids (*Figure 5B–C*).

We next tested whether the extracellular role of eAGR2 requires the presence of the KTEL motif (*Figure 6D*). Indeed, the carboxyl-terminal KTEL domain is required for intracellular AGR2 enzymatic function (*Gupta et al., 2012*). Remarkably, the addition of the human recombinant AGR2-ΔKTEL (40 ng/ml) to the extracellular medium of AGR2-depleted organoids was still able to restore the formation of tumor organoids (*Figure 6D*). Therefore, the KTEL motif is not necessary for eAGR2 extracellular activity.

Next, we investigated whether eAGR2 could play a direct extracellular role through its thioredoxin-like domain (CXXS motif) (*Figure 6E*). To this end, we purified recombinant human AGR2-AXXA, the inactive form of AGR2 thioredoxin-like domain (*Figure 6—figure supplement 1G–J*). AGR2-AXXA mutant, in the ECM of AGR2-depleted organoids, did not restore the formation of tumor organoids (*Figure 6F*). However, the AGR2-AXXA mutant might be improperly folded as displayed in *Figure 6—figure supplement 2*. The sequence C81-S84 is a loop connecting a β-sheet

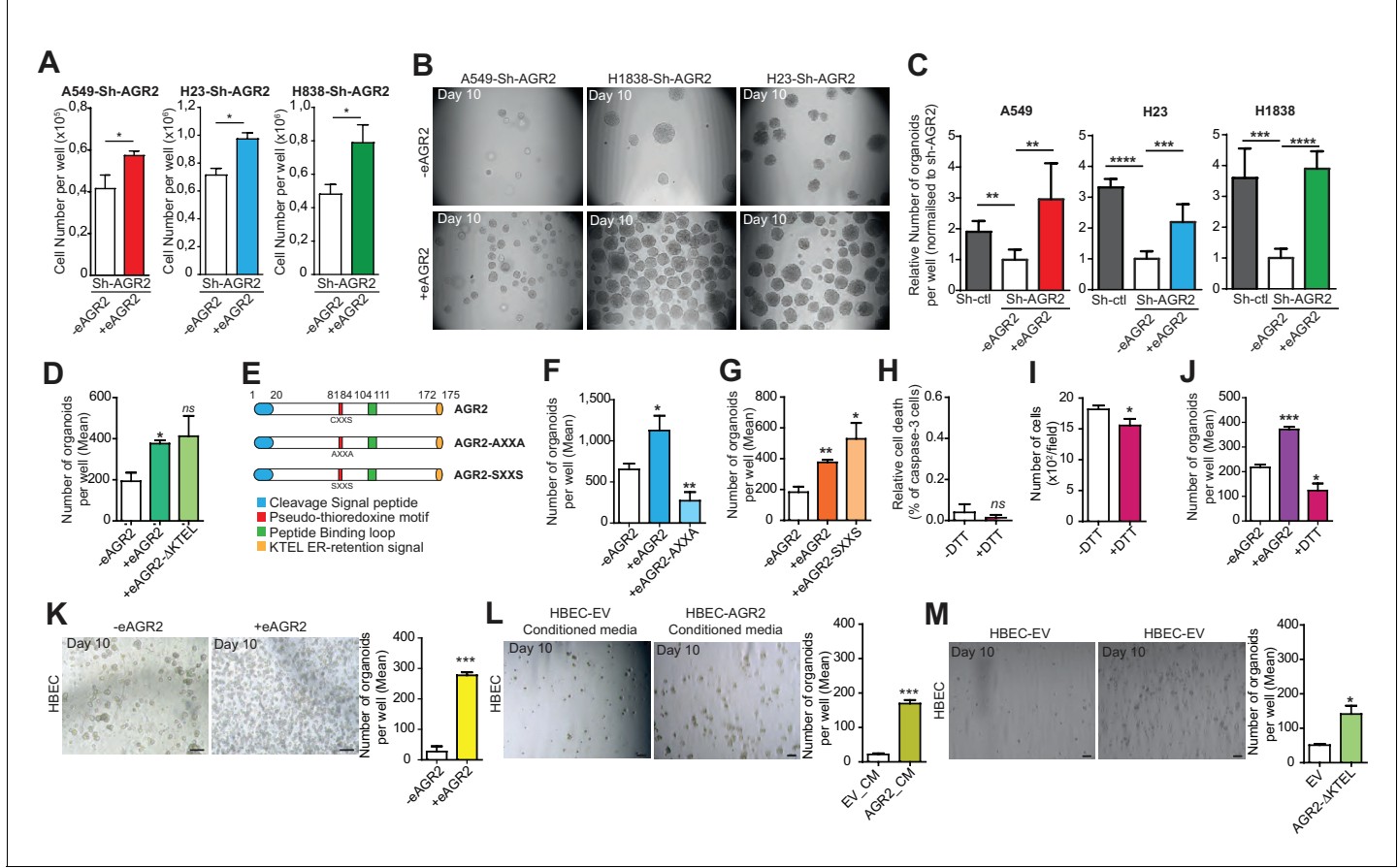

**Figure 6.** Extracellular AGR2 boosts the organoid-initiating frequency. (**A**) Quantification of the numbers of cells per wells following the addition of AGR2 in the extracellular medium (+eAGR2). Data are represented as mean ± SEM of at least 3 independent experiments. *p<0.05. (**B**) Representative brightfield images of tumor organoids grown in 3D from cells harboring Sh-AGR2 in presence (+eAGR2) or not (-eAGR2) of AGR2 in the ECM (n = 3). (**C**) The bar graphs show the mean of organoids per well (mean ± SEM.), after 10 days in presence (+eAGR2) or not (-eAGR2) of AGR2 in the ECM, (n = 3). The P values are relative to untreated cells. **p≤0.01, ***p≤0.001 and ****p≤0.0001. (**D**) The bar graphs show the number of organoids per well in absence (-eAGR2) of AGR2 as compared to cells in presence either of AGR2-AXXA mutant (+eAGR2-AXXA) or AGR2 wt (+eAGR2). (**E**) Schematic representation of AGR2 showing the distribution of AGR2's principal domains and its mutants AXXA and SXXS. (**F**) The bar graphs show the number of organoids per well in absence (-eAGR2) of AGR2 as compared to cells in presence either of AGR2-AXXA mutant (+eAGR2-AXXA) or AGR2 wt (+eAGR2). (**G**) The bar graphs show the number of organoids per well in the absence (-eAGR2) of AGR2 as compared to cells in presence either of AGR2-SXXS mutant (+eAGR2-SXXS) or AGR2 wt (+ eAGR2). (**H**) Quantitation of cell death in H23-Sh-AGR2 depleted cells (Sh-AGR2) in absence (-DTT) or in the presence (+DTT) of DTT. Results are representative of three independent experiments. The cell death rate was determined by the percentage of caspase-3 positive cells. *n.s:* not significant. (**I**) Growth of cells in H23-Sh-AGR2 depleted cells (Sh-AGR2) in absence (-DTT) or in presence of (+DTT) DTT (three independent experiments). Data are mean ± SEM. *p<0.05. (**J**) The bar graphs show the number of H23-Sh-AGR2 depleted organoids per well in absence (- eAGR2) of AGR2 as compared to cells in presence either AGR2 wt (+eAGR2) or DTT (+DTT). Data are mean ± SEM. *p<0.05 and ***p≤0.001. (**K**) Representative images of the organoids formed by HBEC in the presence (+eAGR2) or absence (-eAGR2) of AGR2 in the ECM. The bar graph shows the mean of organoids per well (mean ± SEM., n = 3). The p values are relative to untreated cells. ***p≤0.001. (**L**) Representative images of the organoids formed by HBEC in the presence of conditioned medium from HBEC-vector cells (EV_CM) or in the presence of conditioned medium from HBEC-AGR2 cells (AGR2_CM) in the ECM. The bar graph shows the mean of organoids per well (mean ± SEM., n = 3). The p values are relative to untreated cells. ***p≤0.001. (**M**) Representative images of the organoids formed by HBEC overexpressing AGR2-AXXA (HBEC-AGR2-AXXA) or not (HBEC-EV). The bar graph shows the mean of organoids per well (mean ± SEM., n = 3). *p<0.05.

The following figure supplements are available for figure 6:

**Figure supplement 1.** AGR2 and AGR2 thioredoxin mutant (AGR2-AXXA) purification and activity on organoids formation.

**Figure supplement 2.** Structure model showing the CXXS (81–84) sequence.

strand with an α-helix. The mutation to A could affect the structure, since we thereby loose the key interactions stabilizing the 'folding'/closure of that loop (*Figure 6—figure supplement 2*). S84 forms strong hydrogen bonds to both the carboxylic acid of D79 and the backbone carbonyl of L78 (*Figure 6—figure supplement 2*). Thus, we used another inactive form of AGR2 thioredoxin-like domain, the AGR2-SXXS mutant. In contrast to the AGR2-AXXA mutant, AGR2-SXXS mutant restored the formation of AGR2-depleted tumor organoids (*Figure 6G*). These results show that the single cysteine residue in the AGR2 thioredoxin-like domain is not essential for the formation of tumor organoids. To further assess the redox function of eAGR2 in such process, we treated tumor organoids depleted in AGR2 (H23-Sh-AGR2) with dithiothreitol (DTT) at a concentration of 5µM and no cell death was detected (*Figure 6H*). The addition of DTT to the medium of AGR2-depleted H23 cells neither reversed the cell growth inhibition induced by AGR2 depletion (*Figure 6I*) nor restored the formation of tumor organoids (*Figure 6J*). Thus, these results demonstrate that treating cells with reducing agent cannot mimic the effects of eAGR2.

We next investigated the effects of eAGR2 in non-tumorigenic HBEC (*Figure 6K*). As expected, eAGR2 enhanced organoid growth ten-fold (*Figure 6K*). To determine whether eAGR2 acts through an autocrine/paracrine mechanism, conditioned media from HBEC cells stably infected with empty vector (HBEC-EV) or overexpressing AGR2 (HBEC-AGR2) were added to HBEC organoids (*Figure 6L*). Conditioned media from control (HBEC-EV) did not significantly stimulate the formation of HBEC organoids. In contrast, conditioned media from HBEC-AGR2 stimulates the formation of non-tumorigenic organoids eight-fold (*Figure 6L*). This demonstrated that eAGR2 secreted by organoids and the bacterially expressed human recombinant AGR2 are both active and have the same role on the organoid growth (*Figure 6K–L*). Next, to confirm that eAGR2 achieved its functions through an autocrine/paracrine manner, we explored the functional effect of the AGR2-ΔKTEL protein, which is secreted and not retained by the KDEL receptors of the ER (*Gupta et al., 2012*), on non-tumorigenic organoids (*Figure 6M*). HBECs were infected with either empty vector (HBEC-EV) (*Figure 6M*) or AGR2-ΔKTEL containing vector (HBEC-AGR2-ΔKTEL) (*Figure 6M*). AGR2-ΔKTEL overexpression increased the formation of organoids (*Figure 6M*), thereby confirming that the expression of AGR2 dictates organoid-initiating frequency through an autocrine/paracrine manner. These results demonstrate that eAGR2 extracellular functions were independent of its thioredoxin-like domain and occurred through an autocrine/paracrine manner on organoid-initiating frequency, a phenotype of transformed tumor cells.

## Extracellular AGR2 disrupts epithelial cell polarity and lumen formation

Confocal microscopy analysis of non-tumorigenic HBEC organoids was then used to further evaluate the impact of eAGR2 on organoid architecture. We showed that eAGR2-induced re-distribution of polarization markers using confocal microscopy in HBEC organoids (*Figure 7A–D*). As shown in *Figure 7A and B* (bottom and right panels), the apical marker GM130 was randomly distributed throughout the HBEC organoids in presence of eAGR2 in the ECM, in contrast to its restriction to the apical localization in HBEC organoids in the absence of eAGR2 (*Figure 7A and B* (top and left panels). Fluorescence intensity cross-section profile (*Figure 7B*) revealed that the intensity of GM130 (green) had two side peaks that referred to the control HBEC organoid shell (- eAGR2). In presence of eAGR2, GM130 distributed randomly within the HBEC organoid (*Figure 7B*). Similarly, control HBEC organoids exhibited a lumen lined by the tight junction marker ZO-1 (*Figure 7C and D*; top and left panels), whereas ZO-1 was no longer restricted to the lumen compartment upon eAGR2 exposure (*Figure 7C and D*; bottom and right panels). Thus, eAGR2 in the ECM, resulted in a disruption of apicobasal polarity, of non-tumorigenic organoids. Therefore, we hypothesize that eAGR2 determines the orientation of epithelial polarity and thereby lumen formation. Hence, we next investigated the development of lumen in HBEC organoids, in the absence (- eAGR2) or in the presence (+ eAGR2) in the ECM of eAGR2 (*Figure 7E–G*) using apical F-actin (*Figure 7E,F*). The presence of eAGR2, at the time of plating, gave rise to organoids unable to form a lumen (+ eAGR2) (*Figure 7E–G*) thereby showing that eAGR2 interferes with the formation of hollow organoid. Collectively, these data provide further evidence that eAGR2, in the ECM, induced loss of cell polarity and lumen formation.

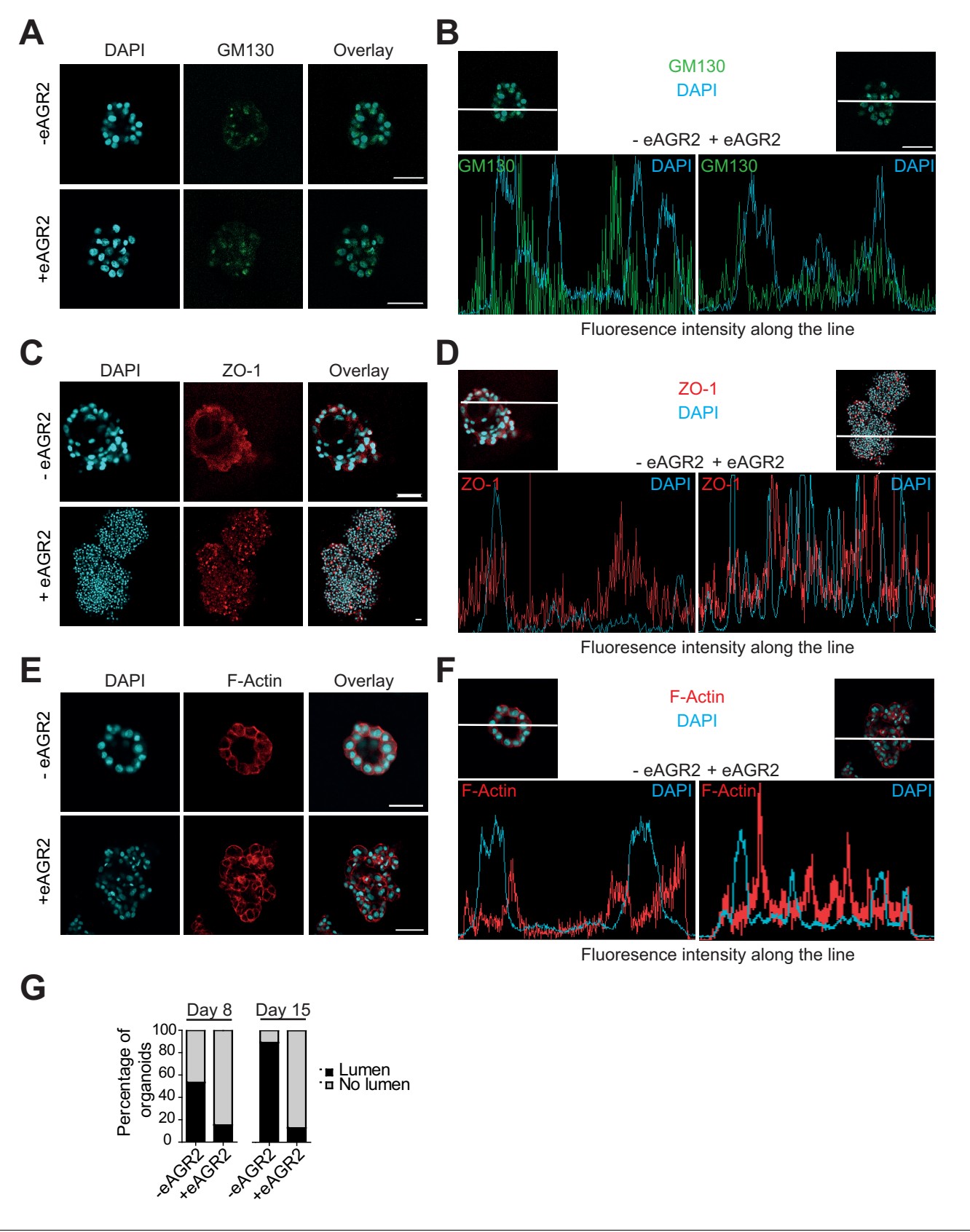

**Figure 7.** Extracellular AGR2 disrupts apico-basal polarity and lumen formation. (**A**, **C**, **E**) HBEC cultured organoids in the presence (+ eAGR2) or absence (-eAGR2) of AGR2 in the ECM, stained for GM130 (**A**), ZO-1 (**B**) and F-Actin (**C**). Scale bars, 50 µm. Note the repositioning or fragmentation of Golgi following the presence of eAGR2. (**B**, **D**, **F**) Fluorescence intensity profile along the white line across the organoid in the presence (+ eAGR2, right panel) or absence (-eAGR2, left panel) of AGR2 in the ECM, stained with DAPI and GM130 (**B**), ZO-1 (**D**) and F-Actin (**F**). (**G**) Quantification of organoids with lumens; n>100 for each condition, (n=3).

## Extracellular AGR2 disrupts epithelial cell-cell adhesion

Confocal microscopy analysis of non-tumorigenic HBEC organoids was then used to further evaluate the impact of eAGR2 on cell-cell adhesion. We investigate the possibility that eAGR2 disrupts tissue organization by affecting cell-cell adhesion. The HBEC organoids were stained with the following cell-cell adhesion markers E-cadherin (*Figure 8A*), β-catenin (*Figure 8B*) and laminin-V (*Figure 8C*). E-cadherin, β-catenin and Laminin-V relocalized from the basal side of the organoids to cell cytoplasm (*Figure 8A–B*). Thus, eAGR2 in the ECM, resulted in a disruption of cell-cell contact, of non-tumorigenic organoids.

## Extracellular AGR2 controls Epithelial–Mesenchymal Transition (EMT)

Since AGR2 has been correlated as potentially involved in EMT (*Mizuuchi et al., 2015*; *Ma et al., 2015*) and that EMT is characterized by a reduction of cell-cell adhesion and loss of apico-basolateral polarity (*Thiery, 2002*), we examined whether eAGR2 might regulate EMT in non-tumorigenic HBEC organoids. To this end, HBEC organoids growth with eAGR2 for 10 days or not, RNA was harvested and analysed using RT-qPCR array. The expression of 84 EMT genes was quantified and presented as a Volcano plot (*Figure 9—figure supplement 1*). The addition of eAGR2 to the organoids ECM significantly modified EMT transcripts such as Wnt5b (~thirteen-fold), MMP3 (~eleven-fold), MAP1B (~ten-fold), MMP9 (~eight-fold), ZEB1 (~three-fold), and VIM (~two-fold) (*Figure 9A*). Changes were also observed in genes related to ECM and cell adhesion, cell growth and proliferation, differentiation and development, migration and mobility, and cytoskeleton (*Figure 9B*). These results were also confirmed at the protein level with the induction of Vimentin (VIM), MMP9 and N-Cadherin (CDH2) expression upon treatment with eAGR2 as well as decreased E-Cadherin (CDH1) expression (*Figure 9C*). The impact of eAGR2 on EMT induction is also reinforced by the relocalization of E--Cadherin and β-catenin from the basal side of the organoids to the cytoplasm (*Figure 8A and B*).

## Extracellular AGR2 promotes invasive structures

Matrix MetalloProteinases (MMP) have been previously reported to play a critical role in stimulating invasion (*Kessenbrock et al., 2010*). Thus, the induction of MMP3 and 9 by eAGR2 let us to hypothesize that eAGR2 promotes invasion. Since invasion of mesenchymal cells is enhanced by stiff ECM (*Paszek and Weaver, 2004*), we then assessed the impact of eAGR2 on the 3D-invasive potential in stiff ECM. These experiments showed that eAGR2 disrupted laminin layers to invasive structures (*Figure 9D*). Hence, we correlate a cell invasion phenotype with the presence of eAGR2 in the ECM of non-tumorigenic organoids.

## Discussion

The present study reveals an important role for eAGR2 in mediating loss of cellular polarity and morphologic transformation of epithelial cells. To our knowledge, this is the first time eAGR2 is characterized in the epithelial tissue morphogenesis and tumorigenesis, and this information will increase our understanding of the biological and physiological properties of AGR2 in this field. Studies in the last decade demonstrated that despite the integrity of its KTEL signal (ER retention motif) (*Gupta et al., 2012*), the ER-resident AGR2 is secreted in various cancers (*Chevet et al., 2013*). AGR2 is retained in the ER through a mechanism involving its KTEL motif and the three KDEL receptors (*Gupta et al., 2012*; *Alanen et al., 2007*). To further support this aspect, we showed that AGR2 KTEL motif is accessible to bind to the KDEL receptors. Furthermore, we demonstrated that eAGR2 secretion is independent of the KTEL sequence and, is not a consequence of cell death or "leaky" release of ER proteins into the extracellular medium. Moreover, the secretory pathway was not dramatically altered upon exposure to eAGR2. Hence, our data show that the production of AGR2 is

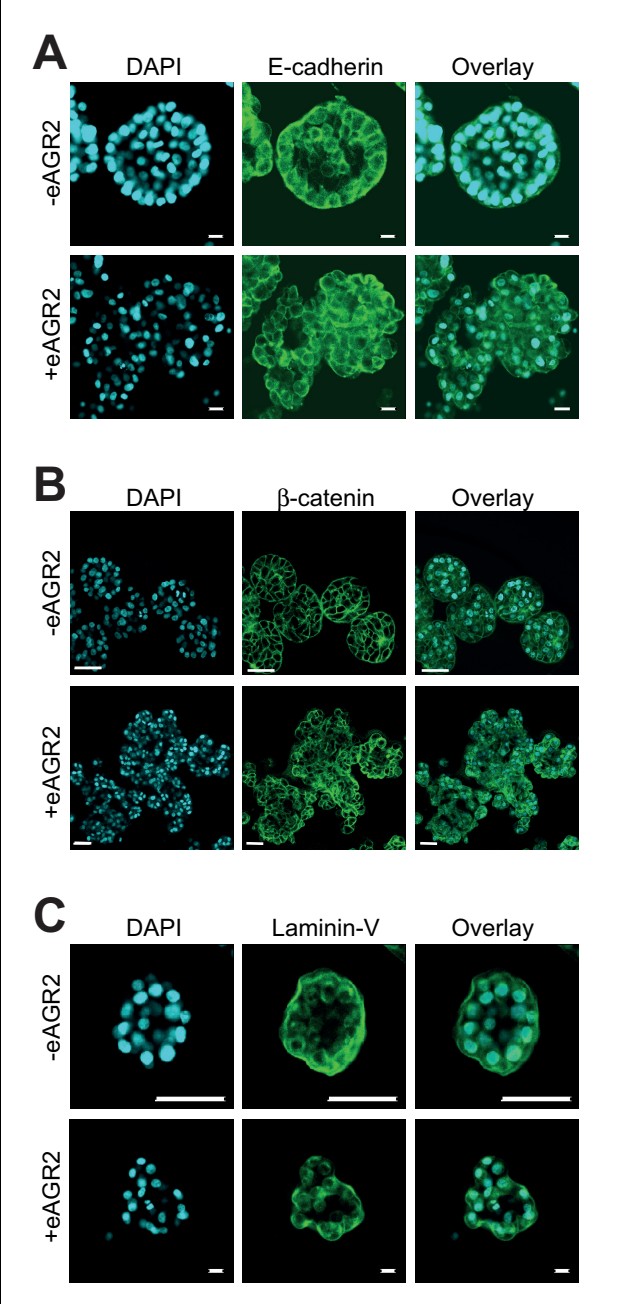

**Figure 8.** Extracellular AGR2 disrupts cell-cell adhesion. Confocal cross-sections of HBEC organoids in the presence (+eAGR2) or absence (-eAGR2) of AGR2 in the ECM stained with E-Cadherin (**A**), β-catenin (**B**), or Laminin-V (**C**), and DAPI (blue) for nucleus. Scale bars, 50 μm.

linked to the amount of secreted eAGR2. Recent studies have suggested that different affinities for the KDEL receptors may impact on subcellular distribution, the latter being dynamic and dependent on conditions such as cell density or growth media (*Alanen et al., 2011*). This could explain how high levels of iAGR2 expression in cancer cells combined with higher cell density and/or altered growth media could impact on the amount and rate of eAGR2 secretion.

We observed that eAGR2, which is undetectable in the ECM of non-tumor organoid, is abundantly secreted by tumor organoids and is present in the ECM. Hence, these data suggest an important role for eAGR2 in controlling ECM functionality, and most likely in the remodeling of tumor

microenvironment. A similar role was described for PDIA3 in remodeling of the ECM during renal fibrosis (*Dihazi et al., 2013*). Moreover, high levels of eAGR2 could directly contribute to the tumor-igenecity of tumor cells. Indeed, AGR2 depletion in cancer cells significantly decreased tumor cell proliferation, suggesting that high levels AGR2 in tumor cells is important for sustaining tumor cell growth. This is consistent with the observation that conditioned media from cells silenced for AGR2 have a reduced ability to stimulate proliferation of pancreatic cancer cells (*Arumugam et al., 2008*). Furthermore, our data show that restoration of eAGR2 in the microenvironment of AGR2-knockdown tumor cells reversed AGR2 depletion-induced cell growth inhibition and thus indicate that the reduced cell proliferation induced by AGR2 depletion is most likely due to the reduction of eAGR2 in the tumor microenvironment.

To further document the biological function of eAGR2, we have used different human adenocarci-noma organoids genetically depleted for AGR2 and showed the decreased ability of cancer cells to form tumor organoids. This is independent of iAGR2 since the reduced tumor organoids formation in AGR2-depleted cells can be restored by addition of exogenous eAGR2. It has been reported that the carboxyl-terminal KTEL domain is specifically required for intracellular AGR2 enzymatic function (*Gupta et al., 2012*). Our data reveal that for the eAGR2 extracellular activity, the KTEL motif is not required. Remarkably,, tumor organoid formation was restored when AGR2-depleted tumor organo-ids were supplemented with AGR2-SXXS (thioredoxin inactive), but not in the presence of reducing agent This result suggests a protein-protein interaction between eAGR2 and ECM proteins and/or proteins involved in the control of cell adhesion and cell-matrix interaction in an AGR2 thioredoxin-like domain independent fashion. The interaction of AGR2 with several proteins involved in such pro-cesses has been described in the literature. Indeed, AGR2 was found to interact with DAG1 (*Fletcher et al., 2003*), LYPD3 (*Alanen et al., 2007*), C4.4A (*Arumugam et al., 2015*), however the exact mechanisms by which AGR2 interacts with those proteins remains unclear and must be further investigated. Nevertheless, our data suggest that this occurs in a thioredoxin-like domain indepen-dent fashion. Therefore, our data demonstrate that eAGR2 drives tumor organoids formation, not by changing the redox nature of microenvironment but through its presence within the ECM. How-ever, the precise mechanisms remain to be further investigated. Our experiments provide the first evidence that eAGR2 has a crucial biological function in the tumorigenesis of cancer cells. Therefore, we hypothesized that eAGR2 could be a pro-oncogenic protein secreted in the ECM.

To confirm that eAGR2 is indeed, an ECM microenvironmental pro-oncogenic protein, we investi-gated the effects of overexpressing AGR2 in human bronchial epithelial non-tumor cells. We found that overexpressing AGR2 enhanced cell proliferation and increased the formation of organoids. Fur-thermore, we demonstrated that the presence of eAGR2 in the ECM was sufficient to impact on cell's autocrine/paracrine signaling to in turn enhance organoid-initiating frequency. Therefore, we investigated the pro-oncogenic mechanisms of eAGR2 in non-tumor organoids to mimic the condi-tions under which eAGR2 acts in vivo. In normal conditions, HBECs form polarized and spherical organoids with hollow lumens, however in the presence of eAGR2 in the ECM, these identical cells form large, non-polarized, undifferentiated organoids without lumen, therefore mimicking tumor organoids. Disruption of cell polarity and tissue organization is reported to be an important event for the initiation and progression of tumorigenesis. Hence, eAGR2 alone is sufficient to confer a tumorigenic phenotype in non-tumor organoids. We next examined the impact of eAGR2 on cell–cell adhesion and the development of invasive structures. We demonstrated that eAGR2: i) pro-motes a cell invasion phenotype characterized by the disruption of basal laminin and, ii) disrupts cell-cell contact characterized by the loss of E-cadherin, β-catenin and Laminin-V at the cell membrane. Adherens junctions represent a powerful invasion suppressor complex in normal epithelial cells (*Pećina-Slaus, 2003*). Disruption of polarized tissue structure and induction of cellular invasion are accompanied by extensive remodeling of the cellular microenvironment (e.g. basement membrane degradation, disruption of cell-ECM interactions). In such context MMPs play critical roles as they stimulate tumorigenesis, cancer cell invasion and metastasis (*Stallings-Mann et al., 2012*; *Arumugam et al., 2008*). This is consistent with induction of MMP3 and MMP9 observed in eAGR2-treated non-tumor organoids. MMP9 may function to activate local growth factors, stimulate angio-genesis in vivo, and degrade the ECM during cell invasion, whereas MMP3 is critical for the cleavage of E-cadherin and is involved in promoting EMT and invasion in epithelial cells (*Xian et al., 2005*; *Fingleton et al., 2001*; *Lochter et al., 1997*). This is a mechanism through which eAGR2 may act to promote cell invasion. Moreover, eAGR2 triggers the induction of Vimentin and N-Cadherin.

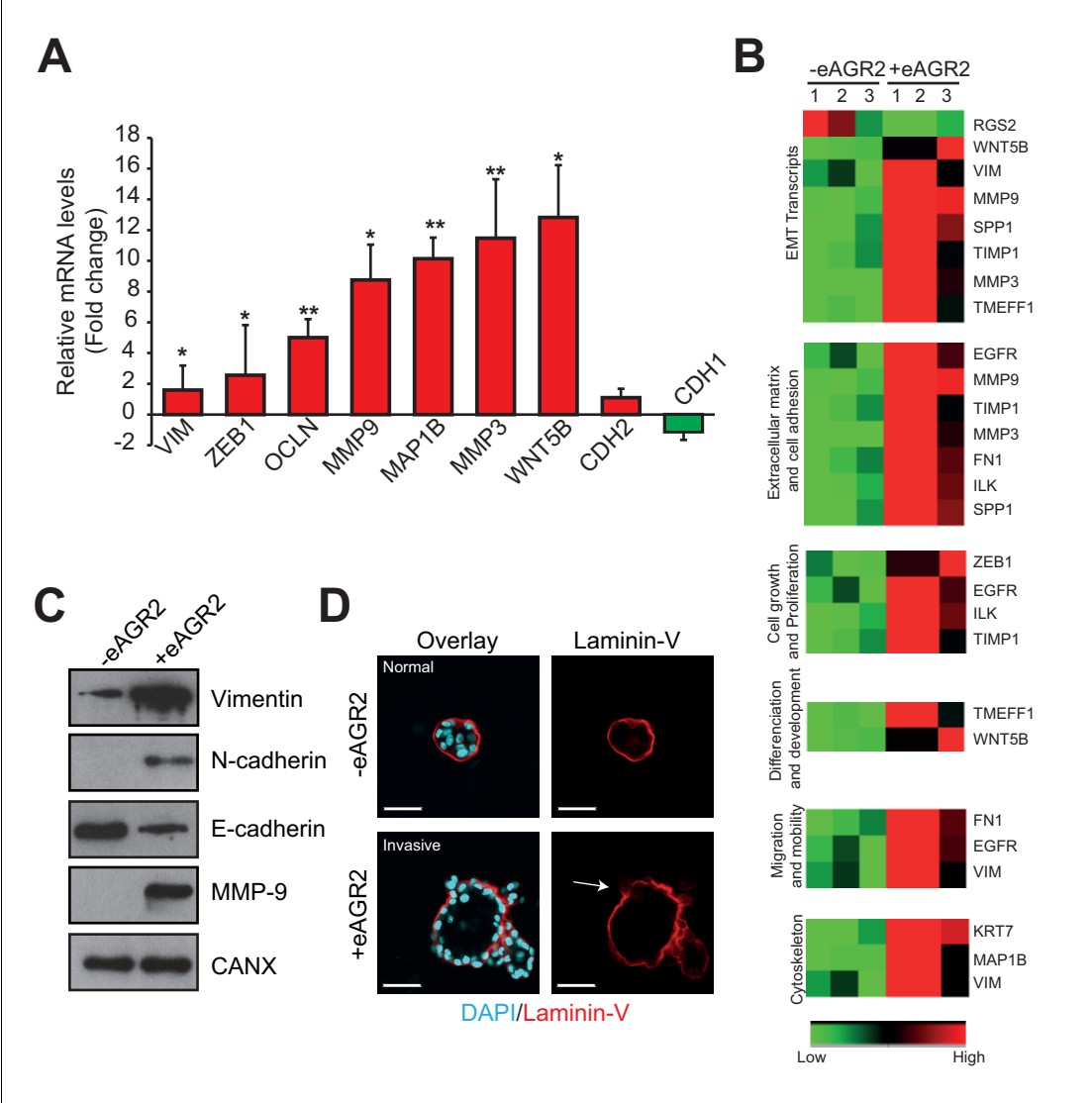

**Figure 9.** Extracellular AGR2 promotes EMT and invasive structures. (**A**) Transcript expression of the EMT transcriptional effectors from the array validated by qRT-PCR. Quantitative PCR levels were normalized to B2M expression. *p≤0.05. **p≤0.01. (**B**) A focused EMT qRT-PCR array was utilized to assess EMT-regulated genes modulated by eAGR2 in HBEC cells. Samples for array data were derived from three independent replicates of the biological experiments. (**C**) Down-regulation of E-Cadherin and up-regulation of Vimentin, N-Cadherin and MMP9 protein concentrations in organoids in presence of AGR2 in the ECM (+ eAGR2) compared to organoids in absence of AGR2 in the ECM (- eAGR2), as analyzed by western blot. Calnexin (CANX) concentration is shown as loading control. One representative experiment (n = 3) is shown. (**D**) Invasion phenotype induced by eAGR2 on matrigel/collagen mixture. Organoids are fixed and stained with anti-Laminin 5 antibody. Arrows show loss of laminin at the basement membrane on invasive structures.

The following figure supplement is available for figure 9:

**Figure supplement 1.** A volcano plot representation of differentially expressed gene in comparison of cultures growing in absence or presence of eAGR2.

Expression of these mesenchymal markers, together with the loss of E-cadherin and β-catenin from cell–cell contacts, as well as the invasive behavior of the organoids suggests that the non-tumorigenic epithelial cells were converted from an epithelial to a mesenchymal phenotype in presence of eAGR2 in their ECM. We also observed that eAGR2 induced the expression of other EMT-related genes. EMT is a hallmark of tumor progression associated with the acquisition of invasive and metastatic features (*Hanahan and Weinberg, 2011*). Although it has been only correlated that AGR2

may stimulate EMT, it is intriguing to find that eAGR2 in ECM is sufficient by itself, to induce EMT in non-tumor epithelial cells. Together, our results indicate that eAGR2 is important for epithelial morphogenesis disruption and metastatic dissemination. Thus, these results provide new insight into the possible mechanisms through which AGR2 may function as a microenvironmental pro-oncogenic protein.

In conclusion, the findings from the present study demonstrate that eAGR2 is a microenvironmental regulator of epithelial tissue architecture, which has a role in the preneoplastic phenotype and thus, contributes to tumorigenicity in epithelial cells. Indeed, eAGR2 present in the ECM of non-tumorigenic organoids is sufficient, by itself, to disrupt cell polarization and organoid formation, thus leading to a disorganized phenotype with invasive properties. In epithelial cells, dynamic remodeling of the ECM is essential for the development and normal tissue homeostasis. However, loss of polarity and uncontrolled ECM remodeling leads to cancer. Together, our results highlight that eAGR2 represents a novel and unexpected microenvironmental signaling in the epithelial morphogenesis and tumorigenesis which might be explored for development of novel therapeutic strategies for epithelial cancer.

## Materials and methods

### Cell culture, AGR2 knockdown and overexpression

Human normal bronchial epithelial cells (HBEC) and lung cancer cells have been previously described (*Fessart et al., 2013*). HBEC (Human Bronchial Epithelial Cells) cells were purchased from Lonza. A549, H1838, H23 and HEK-293 (CRL-1573) cells were purchased from American Type Culture Collection. All cell lines used for this study were tested free of mycoplasma. All cancer cells were cultured in DMEM supplemented with 10% FBS. ShRNA against AGR2 (MSCV-LTRmiR30-PIG retroviral vector [*Arumugam et al., 2008*]) and the corresponding scramble control (Sh-ctl) were a gift from Pr. Lowe AW (Stanford University, California, USA). The MSCV-LTRmiR30-PIG retroviral plasmid was used to produce retroviral particles in the Phoenix Ampho HEK293T cell line. After transduction into A549, H1838 or H23 cells, selection was performed for 3 days with puromycin (2 μg/ml for each cell line). AGR2 concentrations were analyzed quantitatively by western blot and qualitatively by immunofluorescence. For AGR2 overexpression experiments, HBECs were infected with an empty vector pLenti6.3/V5 or a pLenti6.3/V5 vector expressing the coding sequence of human AGR2. Selection was performed for 3 days with blasticidin (5 μg/ml). For AGR2 and AGR2-AXXA recombinant, AGR2 from pLenti6.3/V5 vector was sub-cloned into pET-60-DEST vector. Each construct was verified by DNA sequence analysis. AGR2 mutant cDNAs (C81S, K172D, K172A, K172Stop, T173D named respectively SXXS, DTEL, ATEL, DeltaKTEL and KDEL) (*Figure 5J*) were synthesized and cloned in pGEX-2TK for GST-AGR2 production and pcDNA4.1 for mammalian expression.

### Antibodies and reagents

We used commercial matrix (Matrigel) for the organotypic assays. The sources of the primary antibodies used in these studies were as follows: β-catenin (BD Biosciences), cleaved-caspase 3 (Asp175) (Cell Signaling), Calnexin (Stressgen), GM130 (BD Transduction Laboratories), laminin V (Millipore), mouse AGR2 (Abnova, clone 1C3), rabbit polyclonal AGR2 antibody (Abcam), BiP antibody (Abcam), GAPDH antibody (Millipore), Transferrin antibody (Cell signalling), and O-linked-N-acetylglucosamine antibody (Abcam).

The secondary antibodies used were as follows: Alexa-Fluor-488- and Alexa-Fluor-546-conjugated anti-mouse and rabbit IgGs (Molecular Probes/Invitrogen). Reagents used in the study were diaminophenylindole (DAPI) (Sigma-Aldrich, St Louis, MO), paraformaldehyde (Sigma-Aldrich), DTT (Sigma-Aldrich) and Cycloheximide (Sigma-Aldrich).

### Soft-agar growth assay

Anchorage-independent growth was evaluated as described (*Lleonart et al., 2010*). Briefly, $1 \times 10^5$ cells were plated in DMEM with 3.3% FBS containing 0.35% soft agar in 6-cm plates over a layer of solidified DMEM containing 0.7% soft agar. Medium was added twice a week to maintain humidity. After 4 weeks, colonies were stained with Crystal Violet (0.05% ) for 10 min and counted.

## In vivo tumorigenicity assay

A549-Sh-ctl ($1 \times 10^6$), A549-Sh-AGR2 ($1 \times 10^6$) or A549-Sh-ctl cells ($1 \times 10^6$) and A549-Sh-AGR2 ($1 \times 10^6$) were injected into the tail of male RAG2 -/- gamma mice. Mice were followed weekly, and when tumors in any of the groups (mice injected with Sh-ctl or Sh-AGR2 cells) reached around 1 cm$^3$, determined by palpation (at 2 weeks), all mice were killed, and their lungs were dissected and processed. A total of 44 mice (12 for each cell lines, A549 and H1838) were used for each condition (Sh-ctl and Sh-AGR2). All animal procedures met the European Community Directive guidelines (Agreement B33-522-2/ N°DIR 1322) and were approved by the ethical committee.

## Tissue samples, immunohistochemistry and histopathological analysis

Samples of human lung cancer tissues were obtained from the Haut-Levêque University Hospital (Bordeaux, France) and reviewed by expert pathologist in the field (H. Begueret). The number of human lung samples analyzed and their clinicopathological characteristics are described in *Supplementary file 1A*. These procedures were approved by the Institutional Review Board at Haut-Levêque University Hospital (NFS96900 Certification). Immediately after the surgical resection, tumor specimens were fixed for 24 hr in 10% buffered formaldehyde, dehydrated and routinely embedded in paraffin. Immunohistochemical analyses were performed using 4 µm sections. All staining procedures were performed in an automated immunostainer (Bond-III, Leica Biosystems Newcastle Ltd, Newcastle-Upon-Tyne, U.K) using standard reagents provided by the manufacturer. Briefly, after Bond Epitope Retrieval solution 2 (Leica Biosystems Newcastle Ltd, Newcastle-Upon-Tyne, U.K) antigen retrieval for 20 min, de-paraffinized sections were incubated with the anti-AGR2 (Abnova, clone 1C3) human monoclonal antibody at a 1:100 dilution for 15 min at room temperature. For visualization, the Bond Polymer Refine Detection kit (Leica Biosystems Newcastle Ltd, Newcastle-Upon-Tyne, U.K) was used according to the manufacturer's instructions. Each immunohistochemical run contained an internal positive control (bronchiolar epithelium or type II pneumocytes) and an antibody negative control (buffer, no primary antibody). Sections were visualized with a Nikon-Eclipse501 microscope, and images were acquired using NIS-Elements F 3.00 System (Nikon Digital Sight). Immunohistochemical staining was scored by intensity using the following criteria: 0, no staining; 1, mild staining and 2, dark staining. Quantification was done using ImageJ software and analyzing 20 images (20X) per tumor. Tumors of at least six mice were used for quantification. The number of tumor foci was calculated from whole lung sections with a cut-off of >0.32 mm for tumor nodule diameter.

## Immunofluorescence microscopy

Cells cultured on cover slips were fixed using 4% PFA in PBS for 15 min and then permeabilized with 0.25% Triton X-100. After blocking with 1% BSA and 1% FBS in PBS, cells were incubated with the primary antibody for 1 hr at RT. Cells were then washed and incubated with secondary antibodies (Life Technologies) for 1 hr at RT. DAPI staining was used to visualize nucleus. For three-dimensional culture (3D) organoids, cells were grown in laminin-rich basement membrane growth factor reduced Matrigel (BDBiosciences) (Matrigel) as we previously described (*Fessart et al., 2013*). Confocal analysis was performed using Nikon confocal imaging system. Images were generated and converted to Tiff format.

## Western blotting (WB)

Cells were lysed in buffer A (0.05 M Tris-HCl, 0.15 M NaCl, 1% Triton X-100, pH 7.2) containing protease and phosphatase inhibitors. After centrifugation, proteins in the supernatant were quantified, boiled with Laemmli buffer, resolved by SDS-PAGE and transferred to a nitrocellulose membrane. WB was performed as described previously (*Alanen et al., 2007*). For conditioned medium (CM), cells were grown until confluence, washed and grown in cell culture medium without FBS for 72 hr. CM was collected and concentrated using 30 kDa Centricon filters (Millipore). Protein amounts were assessed by western blot. Mouse AGR2 monoclonal antibody (Abnova, clone 1C3) and rabbit polyclonal Calnexin were used as primary antibodies as described previously (*Alanen et al., 2007*) and peroxidase-conjugated anti-rabbit and anti-mouse Ig (Amersham) as secondary antibodies.

## Stimulation with conditioned media

HBEC cells were seeded in 3D matrigel in the presence of conditioned medium from either HBEC-vector or HBEC-AGR2 cells grown in 2D. Conditioned medium was collected, filtered and centrifuged at 2800 g for 5 min. The medium was changed every forty-eight hours and the cells were incubated for an additional 48 hr. HBEC cells (placed in 3D matrigel) were treated with the conditioned media collected from HBEC-vector cells or HBEC-AGR2, diluted 1:3 with fresh medium. After 48 hr, the medium was changed to fresh diluted 1:3 with the conditioned medium. Ten days later, cell growth in 3D was performed as described previously.

## Detection of secreted AGR2

HEK293T cells were cultured in 6-well plates for 24 hr, washed in OptiMEM (Thermo scientific) and incubated with 500 µL OptiMEM for transfection with wild-type or AGR2 mutants using Fugene 6 (Promega) according to the manufacturer's recommendations. After 24 hr, supernatants were collected, centrifuged at 4500 g for 5 min. Transfected cells were then lysed using 30 mM Tris-HCl pH7.5, 150 mM NaCl and 1% TX100. Supernatants and cell lysates were spotted on nitrocellulose membranes using the Bio-dot mannifold (Bio-Rad France). AGR2 was detected as described above. Dot intensity was quantified using the ImageJ software (imagej.nih.gov/ij) and results were presented as normalized secreted AGR2.

## Cycloheximide pulse-chase analysis

Subconfluent H23 cells were treated with cycloheximide (50 µg/ml) treatment and lysis with lysis buffer (50 mM Tris-HCl, pH 7.4, 150 mM NaCl, 2 mM EDTA, 50 mM NaF, 0.5% NP-40, 1 mM Na3VO4, 2 µg/ml aprotinin, 1 µg/ml PMSF, and 1 µg/ml leupeptin). An AGR2-specific mouse monoclonal antibody was used for immunoprecipitation, followed by immunoblot analysis using AGR2 antibody.

## ECM purification

To generate cell-derived matrices, HBECs were plated in 6-well plates and cultured for 8 days, changing the growth medium every two days. To purify ECM, cells were removed using the published protocol (*Rashid et al., 2012*). Briefly, growth medium was aspirated and cells were washed with PBS. Cell-derived matrices were denuded of cells by dissolving the cell layer first with a solution of 0.5% (v/v) Triton X-100 in PBS and then with 25 mmol/L NH4OH in PBS, for 3 min each, followed by four washes with PBS. This ECM from HBECs in culture remained intact, firmly attached to the entire area of the tissue-culture dish and free of nuclear or cellular debris. Cell-derived matrices were recovered in Laemmli buffer by scraping, resolved by SDS-PAGE and transferred to a nitrocellulose membrane for Western blotting. For immunofluorescence, cells were plated onto coverslips and grown for 8 days before preparing the cell-derived matrices as described above. ECM was fixed with 4% paraformaldehyde and proceeds to immunofluorescence as described previously.

## Statistical analyses

All results were evaluated using GraphPad Prism statistical software package. Different statistical tests were used according to the type of data analyzed (Student's t test, log-rank tests), as is indicated in figure legends. $p \leq 0.05$ was considered statistically significant.

## Reverse transcription (RT)-PCR and real-time quantitative PCR

HBECs in presence (+ eAGR2) or in absence (- eAGR2) of AGR2 were cultured on 3D Matrigel for 8 days and their total RNA was extracted using the RNeasy Mini RNA Isolation Kit (Qiagen). One microgram of total RNA was reverse-transcribed to cDNA by the RT2 First Strand Kit (Qiagen) for each plate. Real-time PCR for RT2 Profiler PCR Array was carried out in a 25 µL solution containing cDNA, 2 × RT2 SYBR Green Mastermix, cDNA synthesis reaction and RNase-free water on an EMT Signaling Pathway RT2 Profiler PCR Array Plate (Qiagen) in a One-Step Plus Real-time PCR system (Applied Biosystems). Real-time RT-PCR profile consists of 10 min of initial activation at 95℃ followed by 40 cycles of 15-s denaturation at 95℃, and 1-min annealing and extension at 60℃ and an association stage including 15-s at 95℃, 1-min at 60℃ and 15-s at 95℃. The relative gene expression data was analyzed by RT2 Profiler PCR Array data analysis v3.5 (Qiagen). Each array per plate

was performed in triplicate, and contained 89 genes including 5 housekeeping genes and 84 genes involved in EMT signaling pathways. The details of these genes are listed in *Supplementary file 1B*.

## Production of recombinant AGR2 protein and AGR2 mutants

Recombinant GST fusion AGR2 proteins were expressed in Escherichia coli (DH5α) and purified as previously described (*Alanen et al., 2007*; *Delom and Chevet, 2006*). Following purification, recombinant proteins were analyzed by SDS-PAGE and quantified by spectrophotometry.

## Acknowledgements

This work was supported by grants from 'La Ligue contre le Cancer' Comités Gironde et Dordogne to DF and Institut National du Cancer (INCa_5869) to CE. FD was supported by 'La Ligue Contre le Cancer' Comités Gironde et Dordogne. We would like to acknowledge G Perot for help in setting up the One-Step Plus Real-time PCR system, Pr. AW Lowe (Stanford University, California, USA) for his kind gift of the MSCV-LTRmiR30-PIG retroviral vector and the corresponding scramble control, the Vectorology platform (Bordeaux, France) for the pVSVG psPAX2 envelope and packaging lentiviral plasmids and the members of the Savineau's laboratory for their critical remarks. The authors would like to thank Dr. R Nookala of Institut Bergonié for the medical writing service.

## Additional information

### Funding

| Funder | Grant reference number | Author |
| --- | --- | --- |
| Ligue Contre le Cancer | | Delphine Fessart<br>Frederic Delom |
| Institut National Du Cancer | INCa_5869 | Eric Chevet |

The funders had no role in study design, data collection and interpretation, or the decision to submit the work for publication.

### Author contributions

DF, Conceived the study, Designed and carried out the experiments, Interpreted the data, Wrote and revised the manuscript, Contributed unpublished essential data or reagents; CD, Carried out the recombinant AGR2 production, Conception and design, Acquisition of data, Analysis and interpretation of data; TA, Cloned the AGR2 mutants and performed the secretion experiments, Analysis and interpretation of data, Drafting or revising the article; LAE, Performed the structural analysis of AGR2, Analysis and interpretation of data, Drafting or revising the article; HB, Carried out immunohistochemistry of human lung cancer patients and mice lung samples, respectively, Analysis and interpretation of data, Drafting or revising the article, Contributed unpublished essential data or reagents; RP, Supervised animal work, Conception and design, Acquisition of data, Drafting or revising the article; CM, Participated to the CHX experiment, Analysis and interpretation of data, Contributed unpublished essential data or reagents; ND-S, Conception and design, Acquisition of data, Analysis and interpretation of data; CL, Carried out the bioinformatics and statistical analyses, Conception and design, Drafting or revising the article; EC, Interpreted the data and revised the manuscript, Contributed unpublished essential data or reagents; FD, Conceived and supervised the study, Interpreted the data and revised the manuscript, Acquisition of data, Contributed unpublished essential data or reagents

### Author ORCIDs

Delphine Fessart, http://orcid.org/0000-0001-7566-5670
Eric Chevet, http://orcid.org/0000-0001-5855-4522
Frederic Delom, http://orcid.org/0000-0002-4600-7633

### Ethics

Human subjects: Samples of human lung cancer tissues were obtained from the Haut-Leveque University Hospital (Bordeaux, France) and reviewed by expert pathologist in the field (H. Begueret).

These procedures were approved by the Institutional Review Board at Haut-Leveque (NFS96900 Certification).

Animal experimentation: All animal procedures met the European Community Directive guidelines (Agreement B33-522-2/ Number DIR 1322) and were approved by the ethical committee from Bordeaux University.

## Additional files

**Supplementary files**

• Supplementary file 1. Table 1A and Table 1B. (A) AGR2 staining and clinicopathological characteristics of human lung cancer tissues samples analysed. (B) Gene Table of EMT Signaling pathway.

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
