## [Decision Letter]

Thank you for submitting your work entitled "Secretion of Protein Disulphide Isomerase AGR2 confers tumorigenic properties" for consideration by *eLife*. Your article has been reviewed by three peer reviewers, and the evaluation has been overseen by Johanna Ivaska as the Reviewing Editor and Sean Morrison as the Senior Editor.

The reviewers have discussed the reviews with one another and the Reviewing Editor has drafted this decision to help you prepare a revised submission.

Summary:

All three reviewers considered the experiment to be well done and they appreciated the main novel observation: recombinant AGR2 is sufficient to confer proneoplastic phenotypes. Nevertheless, they found the current manuscript to be descriptive and concluded that further functional investigation is needed to clarify how the AGR2, especially the active thioredoxin-like domain (CXXS motif), can regulate the formation of tumor organoids, and how can eAGR2 in ECM enhance the organoid growth by ten-fold. It will also be important to address why is the single cysteine in the active site needed? A solid demonstration that eAGR2 is in ECM and not merely a part of the secretome is currently lacking.

Essential revisions:

1) The authors need to define what allows AGR2 to bypass retrieval by KDEL receptors, whether the amount and rate of secretion increase during tumor progression.

2) Related to these points:

What is the final concentration of AGR2 added to the medium? A titration curve is shown in supplementary material, but the concentration used in the experiments is not clearly indicated. Are they in the ranges found in patients?

Also, it would be important to address the following issues:

Is the KTEL still present and accessible in secreted AGR2?

Is it necessary for activity?

Is the T to D replacement in the C terminal tetrapeptide required for secretion?

3) In the Abstract and throughout the manuscript the authors present the eAGR2 as part of the ECM and in this context playing an important role as microenvironment pro-oncogenic regulator, but the experimental data supporting this statement are missing. The authors did not provide any evidence for the eAGR2 as part of ECM. The fact that the eAGR2 was identified in conditioned medium or secretome does not mean that is part of the ECM. The tissue staining did also not confirm the presence of AGR2 in ECM. There was no colocalization with ECM components in stained tumor tissue – this could be easily confirmed by co-staining with antibodies against ECM components. Co-IP could also be an alternative to prove that the eAGR2 is part of the ECM. And since there is no evidence that AGR2 is part of ECM, the data presented and their interpretation are less convincing.

4) To further clarify the mechanism, the organoid system can be exploited to address the following:

A) Recombinant AGR2 protein can mimic some of the effects on cell outcomes and an AXXA mutant is used as a negative control. The AXXA mutant might be unfolded or denatured and a better control for the CXXS mutant would be a SXXS mutant, not a AXXA mutant. Could this more subtle AGR2 mutant protein be purified and accessed? For example, the SXXS mutant might be active, meaning a protein-protein interaction (PPI), not redox, is driving this event. Can AGR2 function be replaced by chemical reductants like DTT? If so, then redox could be important. Perhaps both a PPI and redox are driving the signalling events. This can be addressed experimentally.

B) If authentic AGR2 can be detected in seasoned media, can this replace recombinant bacterially expressed AGR2 and/or can anti-AGR2 antibodies neutralize this effect?

---

## [Author Response]

Essential revisions:

1) The authors need to define what allows AGR2 to bypass retrieval by KDEL receptors, whether the amount and rate of secretion increase during tumor progression.

To address this point, we have added new experimental data with AGR2 mutants which are presented in Figure 5 of the revised manuscript. As such, secretion experiments (Figure 5) and cycloheximide pulse-chase (Figure 5) are now shown in the manuscript. This has allowed us to better define the mechanisms by which AGR2 could bypass KDEL receptors and be secreted in the extracellular medium.

The description of the results presented in Figure 5 can be found in the Results section of the revised manuscript, “To further correlate the amount of AGR2 produced by tumor organoids to the capacity of cancer cells to secrete eAGR2, we transiently blocked protein synthesis using a cycloheximide (CHX) pulse-chase approach and evaluated the amounts of iAGR2 and eAGR2 in tumor organoids. […] These results demonstrate that the iAGR2 expression increase in tumor organoids was causal to the secretion of eAGR2.”; and “To explore the functional significance of the KTEL motif for the AGR2 secretion, HEK-293T cells which do not secrete AGR2, were used (Figure 5). […] Therefore, none of our different KTEL motif mutants displayed a specific intracellular localization and all were secreted.”

Moreover, we have added a section on this point in the revised Discussion as follows: “Studies in the last decade demonstrated that despite the integrity of its KTEL signal (ER retention motif) (Gupta, Dong, and Lowe 2012), the ER-resident AGR2 is secreted in various cancers (Chevet et al. 2013). […] This could explain how high levels of iAGR2 expression in cancer cells combined with higher cell density and/or altered growth media could impact on the amount and rate of eAGR2 secretion.”

*2) Related to these points:*

What is the final concentration of AGR2 added to the medium? A titration curve is shown in supplementary material, but the concentration used in the experiments is not clearly indicated. Are they in the ranges found in patients?

The final concentration of AGR2 added to the medium is 40 ng/ml. This is now included in the Results section of the revised manuscript: “Remarkably, the addition of recombinant human AGR2 (40 ng/ml) (Figure 6—figure supplement 1) to the medium of AGR2-depleted tumor cell lines reversed the cell growth inhibition induced by AGR2 depletion (Figure 6).”

Moreover, the concentration of eAGR2 that we used for our experiments is in the range of eAGR2 levels of present in the serum from cancer patients. This information has now been added in the Results section: “Given that eAGR2 is secreted by tumor organoids (Figure 5) and that AGR2 has been reported in the serum of cancer patients with an average level of eAGR2 ranging from 0.5 to 20 ng/ml (Chen et al. 2010), we assessed whether eAGR2 might exhibit specific extracellular functions.”

*Also, it would be important to address the following issues:*

*Is the KTEL still present and accessible in secreted AGR2?*

This is an interesting point raised by reviewers. To address this issue, we have used structural data available to visualize the structure of the KTEL domain (Figure 5 of the revised manuscript). This structural analysis (performed by Pr. Leif Eriksson (University Göteborg, Sweden), who is now listed as a co-author in the revised manuscript) led us to the conclusion that the KTEL motif is still present and accessible in secreted AGR2. This is now mentioned in the Results section of the revised manuscript, as follows: “This questioned the accessibility of the ER-retention motif present on AGR2, namely the KTEL sequence. […] Thus, it appears that the KTEL motif really stands out of the core protein and is accessible to bind to its receptors.”

*Is it necessary for activity?*

To address this issue and as suggested by reviewers, we used the recombinant AGR2-△KTEL that lacks the KTEL motif and found that this mutant keeps its ability to stimulate organoid growth. Hence, the KTEL motif is not necessary for its activity.

These results are now presented in the revised Figure 6 and in the Results section as follows: “We next tested whether the extracellular role of eAGR2 requires the presence of the KTEL motif (Figure 6). […] Therefore, the KTEL motif is not necessary for eAGR2 extracellular activity.”

Moreover, we have also discussed these data in the Discussion section of the revised manuscript, as follows: “It has been reported that the carboxyl-terminal KTEL domain is specifically required for intracellular AGR2 enzymatic function (Gupta, Dong, and Lowe 2012). Our data reveal that for the eAGR2 extracellular activity, the KTEL motif is not required.”

Is the T to D replacement in the C terminal tetrapeptide required for secretion?

As suggested by the reviewers, we have measured the secretion capacity of AGR2-KTEL as compared to AGR2-KDEL and two others KDEL mutants (K to A or K to D). These experiments were performed by Dr. Tony Avril (Inserm, Rennes, France), who is now listed as a co-author in the revised manuscript. Our data show that the replacements of T to D, K to A or K to D in the carboxyl-terminal KTEL sequence still allowed AGR2 secretion.

These new results are included in Figure 5 of the revised manuscript and described in the Results section as follows: “To explore the functional significance of the KTEL motif for the AGR2 secretion, HEK-293T cells which do not secrete AGR2, were used (Figure 5). […] Therefore, none of our different KTEL motif mutants displayed a specific intracellular localization and all were secreted.”

*3) In the Abstract and throughout the manuscript the authors present the eAGR2 as part of the ECM and in this context playing an important role as microenvironment pro-oncogenic regulator, but the experimental data supporting this statement are missing. The authors did not provide any evidence for the eAGR2 as part of ECM. The fact that the eAGR2 was identified in conditioned medium or secretome does not mean that is part of the ECM. The tissue staining did also not confirm the presence of AGR2 in ECM. There was no colocalization with ECM components in stained tumor tissue – this could be easily confirmed by co-staining with antibodies against ECM components. Co-IP could also be an alternative to prove that the eAGR2 is part of the ECM. And since there is no evidence that AGR2 is part of ECM, the data presented and their interpretation are less convincing.*

We agree with reviewers that the proof that eAGR2 is part of the ECM needs to be established. As a consequence, new experimental data were added in the revised manuscript demonstrating that eAGR2 interacts with ECM proteins. To this end, two approaches were used including i) an ECM purification strategy and ii) immunofluorescence microscopy.

The method is now described in the revised Materials and methods section of the revised manuscript as follows: “To generate cell-derived matrices, HBECs were plated in 6-well plates and cultured for 8 days, changing the growth medium every two days. […] ECM was fixed with 4% paraformaldehyde and proceeds to immunofluorescence as described previously.”

The corresponding results are presented in the revised Figure 5. We have also added a new paragraph “The secreted eAGR2 interacts with extracellular matrix (ECM)”in the Results section: “Several secreted PDIs were shown to interact with ECM proteins (Dihazi et al. 2013, Ilani et al. 2013). […] As expected, immunofluorescence staining for AGR2 revealed the presence of eAGR2 in the ECM of HBEC-AGR2 cells (Figure 5, lower panels). These results demonstrate the presence of eAGR2 in the ECM.”

We also added a paragraph about the implications of these data in the Discussion section of the revised manuscript as follows:“We observed that eAGR2, which is undetectable in the ECM of non-tumor organoid, is abundantly secreted by tumor organoids and is present in the ECM. Hence, these data suggest an important role for eAGR2 in controlling ECM functionality, and most likely in the remodeling of tumor microenvironment. A similar role was described for PDIA3 in remodeling of the ECM during renal fibrosis (Dihazi et al. 2013).”

*4) To further clarify the mechanism, the organoid system can be exploited to address the following:*

*A) Recombinant AGR2 protein can mimic some of the effects on cell outcomes and an AXXA mutant is used as a negative control. The AXXA mutant might be unfolded or denatured and a better control for the CXXS mutant would be a SXXS mutant, not a AXXA mutant. Could this more subtle AGR2 mutant protein be purified and accessed? For example, the SXXS mutant might be active, meaning a protein-protein interaction (PPI), not redox, is driving this event.*

As suggested by reviewers, the AGR2-SXXS mutant was purified and tested on organoids from H23-Sh-AGR2 cells and compared to the effects yielded by wild type AGR2 (eAGR2). The thioredoxin active eAGR2 mutant restored the formation of tumor organoids. The corresponding results are now presented in revised Figure 6 and in the Results section of the revised manuscript as follows: “However, the AGR2-AXXA mutant might be improperly folded as displayed in Figure 6—figure supplement 2. […] These results show that the single cysteine residue in the AGR2 thioredoxin-like domain is not essential for the formation of tumor organoids.”

We also discussed these data in the Discussion section of the revised manuscript as follows:“Moreover, tumor organoid formation was restored when AGR2-depleted tumor organoids were supplemented with AGR2-SXXS (thioredoxin inactive), but not in the presence of reducing agent, suggesting a protein-protein interaction between eAGR2 and ECM proteins.”

*Can AGR2 function be replaced by chemical reductants like DTT? If so, then redox could be important. Perhaps both a PPI and redox are driving the signalling events. This can be addressed experimentally.*

To address whether the redox function of AGR2 is important for its extracellular functions we assessed the effect of a chemical reductant (DTT), used at a final concentration of 5 µM, on organoid formation. We found that the addition of DTT to AGR2-depleted H23 organoids did not restore the formation of tumor organoids. The corresponding results are presented in revised Figure 6 and described in the Results section of the revised manuscript, as follows: “To further assess the redox function of eAGR2 in such process, we treated tumor organoids depleted in AGR2 (H23-Sh-AGR2) with dithiothreitol (DTT) at a concentration of 5µM and no cell death was detected (Figure 6). The addition of DTT to the medium of AGR2-depleted H23 cells neither reversed the cell growth inhibition induced by AGR2 depletion (Figure 6) nor restored the formation of tumor organoids (Figure 6). Thus, these results demonstrate that treating cells with reducing agent cannot mimic the effects of eAGR2.”

This result is also discussed in the Discussion section of the revised manuscript, as follows:“Moreover, tumor organoid formation was restored when AGR2-depleted tumor organoids were supplemented with AGR2-SXXS (thioredoxin inactive), but not in the presence of reducing agent, suggesting a protein-protein interaction between eAGR2 and ECM proteins. Therefore, our data demonstrate that eAGR2 drives tumor organoids formation, not by changing the redox nature of microenvironment but through its presence within the ECM.”

B) If authentic AGR2 can be detected in seasoned media, can this replace recombinant bacterially expressed AGR2 and/or can anti-AGR2 antibodies neutralize this effect?

As suggested by reviewers, the ability of conditioned medium to replace bacterial recombinant on 3D cell growth in a paracrine/autocrine manner was examined and presented in in revised Figure 6. Our results show that eAGR2 secreted by organoids in the conditioned medium is active and can replace the bacterial recombinant AGR2. This experiment is described in the Materials and methods section of the revised manuscript, as follows: “Stimulation with conditioned media: HBEC cells were seeded in 3D matrigel in the presence of conditioned medium from either HBEC-vector or HBEC-AGR2 cells grown in 2D. […] After 48 hours, the medium was changed to fresh diluted 1:3 with the conditioned medium. Ten days later, cell growth in 3D was performed as described previously.”

The results are also described in the Results section of the revised manuscript: “To determine whether eAGR2 acts through an autocrine/paracrine mechanism, conditioned media from HBEC cells stably infected with empty vector (HBEC-EV) or overexpressing AGR2 (HBEC-AGR2) were added to HBEC organoids (Figure 6). […] This demonstrated that eAGR2 secreted by organoids and the bacterially expressed human recombinant AGR2 are both active and have the same role on the organoid growth (Figure 6).”